# Surprising features of nuclear receptor interaction networks revealed by live-cell single-molecule imaging

**Liza Dahal[1,2]\*, Thomas GW Graham[1,2], Gina M Dailey[1], Alec Heckert[3], Robert Tjian[1,2], Xavier Darzacq[1]\***

[1]Department of Molecular and Cell Biology, Berkeley, United States; [2]Howard Hughes Medical Institute, University of California, Berkeley, United States; [3]Eikon Therapeutics Inc, Hayward, California, Berkeley, United States

## eLife Assessment

This **important** study provides data that challenges the standard model that binding of Type 2 Nuclear Receptors to chromatin is limited by the available pool of their common heterodimerization partner Retinoid X Receptor. The evidence supporting the conclusions is **compelling**, utilizing state-of-the-art single-molecule microscopy. This work will be of broad interest to cell biologists who wish to determine limiting factors in gene regulatory networks.

**\*For correspondence:**
dahal.liza472@gmail.com (LD);
darzacq@berkeley.edu (XD)

**Abstract** Type II nuclear receptors (T2NRs) require heterodimerization with a common partner, the retinoid X receptor (RXR), to bind cognate DNA recognition sites in chromatin. Based on previous biochemical and overexpression studies, binding of T2NRs to chromatin is proposed to be regulated by competition for a limiting pool of the core RXR subunit. However, this mechanism has not yet been tested for endogenous proteins in live cells. Using single-molecule tracking (SMT) and proximity-assisted photoactivation (PAPA), we monitored interactions between endogenously tagged RXR and retinoic acid receptor (RAR) in live cells. Unexpectedly, we find that higher expression of RAR, but not RXR, increases heterodimerization and chromatin binding in U2OS cells. This surprising finding indicates the limiting factor is not RXR but likely its cadre of obligate dimer binding partners. SMT and PAPA thus provide a direct way to probe which components are functionally limiting within a complex TF interaction network providing new insights into mechanisms of gene regulation in vivo with implications for drug development targeting nuclear receptors.

## Introduction

Complex intersecting regulatory networks govern critical transcriptional programs to drive various cellular processes in eukaryotes. These networks involve multiple transcription factors (TFs) binding shared cis-regulatory elements to elicit coordinated gene expression (*Gerstein et al., 2012*; *Pan et al., 2009*; *Reményi et al., 2004*). A distinct layer of combinatorial logic occurs at the level of specific TF-TF interactions, which can direct TFs to distinct genomic sites. Thus, fine-tuning of TF-TF interactions can modulate TF-gene interactions to orchestrate differential gene expression. Often, dimerization between different members of a protein family can generate TF-TF combinations with distinct regulatory properties resulting in functional diversity and specificity (*Nandagopal et al., 2022*; *Puig-Barbé et al., 2023*). For example, E-box TFs of the basic helix-loop-helix (bHLH) family such as MYC/MAD share a dimerization partner MAX. Several studies have shown that switching of

heterocomplexes between MYC/MAX and MAD/MAX results in differential regulation of genes and cell fate (*Amati et al., 1993*; *Bouchard et al., 2001*; *Hurlin and Huang, 2006*; *Xu et al., 2001*).

Type II nuclear receptors (T2NRs) present another classic example of such a dimerization network wherein the distribution of heterodimeric species regulates gene expression (*Bwayi et al., 2022*; *Chan and Wells, 2009*; *Chen et al., 2018*; *Wang et al., 2017*). T2NRs constitute an extensive group of basic leucine zipper TFs that share a common modular structure composed of a well-conserved ligand binding domain (LBD) that mediates heterodimerization with their obligate partner retinoid X receptor (RXR) and a highly conserved DNA binding domain that recognizes consensus sequences termed direct response elements (*Evans and Mangelsdorf, 2014*). Unlike Type I nuclear receptors, T2NRs do not depend on ligand binding for DNA engagement. Rather, binding of ligands to the LBD of chromatin- associated T2NR heterodimers results in eviction of co-repressors and recruitment of co-activators (*Evans and Mangelsdorf, 2014*; *McKenna and O'Malley, 2002*; *Figure 1A*).

As the common obligate partner of many other TFs, MAX and RXR are thought to act as the 'core' for their respective dimerization networks. It has been postulated that availability of such core TFs is likely to be limited in cells, resulting in competition between the various partner TFs involved in the network (*Chan and Wells, 2009*; *Walker et al., 2005*). Indeed, early studies using purified proteins revealed that MAD and MYC compete for binding to MAX with equal affinities, and reduced complex formation was seen for either heterodimer with increasing amounts of a competing partner (*Ayer et al., 1993*; *Baudino and Cleveland, 2001*). Similarly, the T2NRs liver X receptor (LXR) and peroxisome proliferator-activated receptor (PPAR) were observed in vitro to have reduced binding to their respective response elements in the presence of competing T2NRs (*Ide et al., 2003*; *Matsusue et al., 2006*; *Yoshikawa et al., 2003*).

Whether such in vitro systems would capture the complexity and competitive dynamics at play in live cells has remained an unresolved issue. Moreover, to date, in vivo studies of competitive heterodimerization networks have not examined TFs at endogenous expression levels and thus may not accurately recapitulate their dynamic interactions with each other and with chromatin (*Fadel et al., 2020*; *Grinberg et al., 2004*) or do not account for expression levels of relevant TFs in individual cells due to the population averaging nature of most such studies (*Rehó et al., 2023*). To overcome these potential shortcomings, we employed fast single-molecule tracking (fSMT) (*Boka et al., 2021*; *Dahal et al., 2023*; *Elf et al., 2007*; *Hansen et al., 2018*; *Hansen et al., 2017*) and its newly developed complement, proximity-assisted photoactivation (PAPA-SMT) (*Graham et al., 2022*) to test the effects of varying the stoichiometry of a core TF (RXRα) and its partner TF (RARα). We find that, contrary to expectations, the core component in cancer cells is not limiting but rather is in sufficient excess to accommodate more RARα even in the presence of many other endogenous partner T2NRs.

## Results

### Live-cell SMT of KI Halo-tagged RAR and RXR

As an initial test case for studying T2NR interactions, we focused on the heterodimeric partners RARα and RXRα, which are endogenously expressed alongside various other T2NRs in U2OS cells, a well-established cancer cell line for SMT (*Hansen et al., 2017*; *McSwiggen et al., 2019*; *Figure 1A*, *Appendix 2—table 1*). Using CRISPR/Cas9-mediated genome editing, we generated clonal lines with homozygous knock-in (KI) of HaloTag at the N-terminus of RXRα and the C-terminus of RARα (*Heckert et al., 2022b*; *Los et al., 2008*; *Figure 1—figure supplement 1A*). Western blotting confirmed that RARα and RXRα were tagged appropriately and expressed at similar levels to the untagged proteins (*Figure 1B*), while co-immunoprecipitation (co-IP) experiments verified that Halo-tagged RARα and RXRα heterodimerize normally as expected (*Figure 1—figure supplement 1B*). In addition, we confirmed using luciferase assays that the RAR ligand, all-trans retinoic acid (atRA), activated retinoic acid responsive element (RARE)-driven gene expression in wild-type and homozygously edited clones, confirming the normal transactivation function of the tagged proteins (*Figure 1—figure supplement 1C*). Confocal live-cell imaging of cells stained with Janelia Fluor X 549 (JFX549) Halo ligand displayed normal nuclear localization of both Halo-tagged RARα and RXRα (*Figure 1—figure supplement 1D*).

To evaluate how RARα and RXRα explore the nuclear environment and interact with chromatin, we used fSMT with a recently developed Bayesian analysis method, SASPT (*Heckert et al., 2022b*), to infer the underlying distribution of diffusion coefficients within the molecular population, yielding

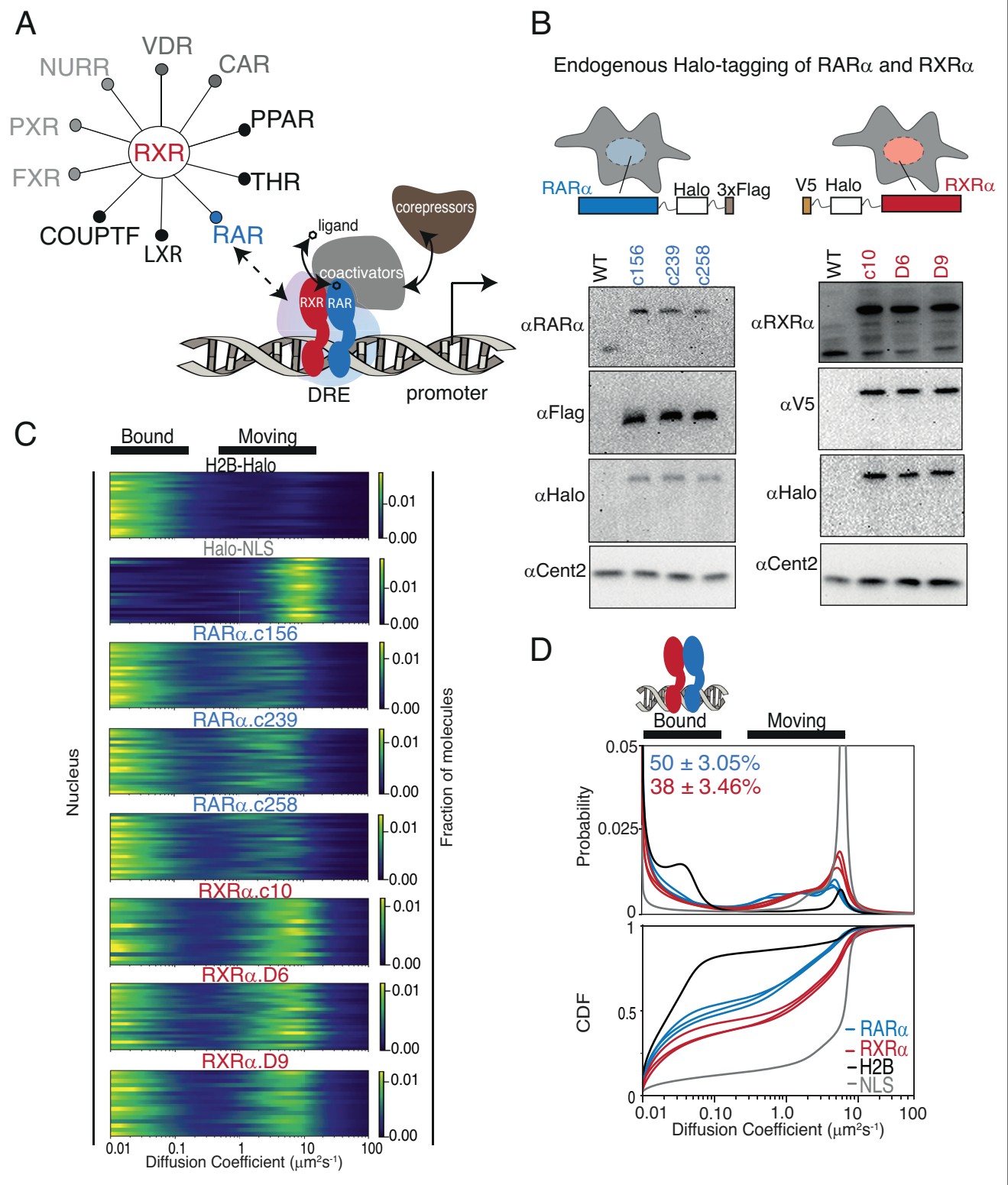

**Figure 1.** Endogenous Halo-tagging of RARa and RXRa to characterize their diffusive behavior. (**A**) Schematic showing Type II nuclear receptors (T2NRs) like RXR-RAR bind direct response elements (DREs) as heterodimers to activate or repress transcription by recruiting co-activators (in presence of ligand) or co-repressors (in absence of ligand). A competitive interaction network between the obligate heterodimeric partner RXR with other T2NRs acts as a complex regulatory node for gene expression. Mechanistic features of protein-protein interaction within this regulatory node and its effect on chromatin binding in live cells are yet to be explored. (**B**) Cartoon showing Halo-tagging scheme of RARa and RXRa along with western blots of U2OS wild-type

*Figure 1 continued on next page*

*Figure 1 continued*

(WT) and knock-in (KI) RARa (left) and RXRa (right) homozygous clones. (**C**) Fast single-molecule tracking (fSMT). Likelihood of diffusion coefficients based on model of Brownian diffusion with normally distributed localization error for H2B-Halo (black), Halo-NLS (gray), RARa clones (blue), and RXRa clones (red) with black lines on top of the figure illustrating bound and moving populations. Each line represents a nucleus. (**D**) Diffusive spectra, probability density function (top), and cumulative distribution function (CDF) (bottom) - with drawing illustrating bound states as heterodimers of RARa and RXRa bound to chromatin.

The online version of this article includes the following source data and figure supplement(s) for figure 1:

**Source data 1.** Includes one PDF file explaining original western blots probing RAR clones with anti-RAR and anti-Flag along with relevant loading controls for *Figure 1B* and *Figure 1—figure supplement 2A*, which are cropped and full-length versions respectively, of the same gels.

**Source data 2.** Includes one PDF file explaining original western blots probing RAR clones with anti-Halo along with relevant loading controls for *Figure 1B* and *Figure 1—figure supplement 2A*, which are cropped and full-length versions respectively, of the same gels.

**Source data 3.** Includes one PDF file explaining original western blots probing RXR clones with anti-RXR (C-term) and anti-RXR (N-term) along with relevant loading controls for *Figure 1B* and *Figure 1—figure supplement 2B*, which are cropped and full-length versions respectively, of the same gels.

**Source data 4.** Includes one PDF file explaining original western blots probing RXR clones with anti-Halo and anti-V5 along with relevant loading controls for *Figure 1*, *Figure 1—figure supplement 2B*, which are cropped and full-length versions respectively, of the same gels.

**Source data 5.** Includes six raw images (tif) of western blots probing RAR clones with anti-RAR and anti-Flag along with relevant loading controls, displayed in *Figure 1B* and *Figure 1—figure supplement 2A*.

**Source data 6.** Includes four raw images (tif) of western blots probing RAR clones with anti-Halo along with relevant loading controls, displayed in *Figure 1B* and *Figure 1—figure supplement 2A*.

**Source data 7.** Includes six raw images (tif) of western blots probing RXR clones with anti-RXR (C-term) and anti-RXR (N-term) along with relevant loading controls, displayed in *Figure 1B* and *Figure 1—figure supplement 2B*.

**Source data 8.** Includes six raw images (tif) of western blots probing RXR clones with anti-Halo and anti-V5 along with relevant loading controls, displayed in *Figure 1B* and *Figure 1—figure supplement 2B*.

**Figure supplement 1.** Verification of endogenous tagging and single-molecule tracking (SMT) analysis.

**Figure supplement 1—source data 1.** Includes one file (pdf) containing and explaining each of the raw images shown in *Figure 1—figure supplement 1B*.

**Figure supplement 1—source data 2.** Includes eight raw images (tif) for co-immunoprecipitation (co-IP) western blots displayed in *Figure 1—figure supplement 1B*.

**Figure supplement 2.** Full-length western blots of wild-type (WT) U2OS cells and homozygously Halo-tagged RARα and RXRα knock-in (KI) clones.

**Figure supplement 3.** Diffusion behavior of endogenous RARα and RXRα with and without all-trans retinoic acid (atRA) treatment.

**Figure supplement 4.** Diffusion coefficient and frame cut-off analysis in single-molecule tracking (SMT) data.

a 'diffusion spectrum' (*Figure 1C and D*). From these diffusion spectra we can extract quantitative parameters of subpopulations (peaks) such as mean diffusion coefficients and fractional occupancy, allowing us to measure the chromatin-bound fraction ($f_{bound}$) in live cells (*Figure 1C and D*). To benchmark our SMT measurements, we first compared the diffusion spectra of RARα-Halo and Halo-RXRα to that of H2B-Halo (which as expected is largely chromatin-bound) and Halo-3×NLS (which is mostly unbound). We observed that RARα and RXRα both exhibit a clearly separated slow diffusing population ($<0.15$ µm²/s) along with a faster mobile population (1–10 µm²/s) (*Figure 1C*). The former we classify as 'bound' since it represents molecules diffusing at a rate indistinguishable from that of Halo-H2B (chromatin motion). The proportion of molecules diffusing at $<0.15$ µm²/s is henceforth designated as $f_{bound}$. In comparison with Halo H2B ($f_{bound} = 75.5 \pm 0.7\%$) and Halo-NLS ($f_{bound} = 10 \pm 0.9\%$), RARα and RXRα have intermediate levels of chromatin binding ($f_{bound}$ of $50 \pm 3.0\%$ and $38 \pm 3.5\%$, respectively) (*Figure 1D*). The $f_{bound}$ was reproducible between three clonal cell lines of RARα ($49 \pm 1\%$, $47 \pm 1\%$, $53 \pm 1\%$) and RXRα ($36 \pm 1\%$, $36 \pm 1\%$, $42 \pm 1\%$). Finally, although recent studies have reported an increase in chromatin interaction upon agonist treatment (*Brazda et al., 2014*; *Brazda et al., 2011*; *Rehó et al., 2023*; *Rehó et al., 2020*), we did not observe a significant change in $f_{bound}$ of RARα and RXRα upon atRA treatment, nor did atRA treatment appear to alter the fast-moving populations of RARα and RXRα (*Figure 1—figure supplement 3A and B*). These results are consistent with the classic model in which dimerization and chromatin binding of T2NRs are ligand independent.

## Chromatin binding of RARα and RXRα can be saturated in live cells

Chromatin binding by an individual TF within a dimerization network is predicted to be sensitive to its expression level (*Klumpe et al., 2023*). To test how the expression levels of RARα and RXRα affect $f_{bound}$, we overexpressed Halo fusions from stably integrated transgenes in U2OS cells (*Figure 2A*). After confirming transgene expression by western blotting (*Figure 2A*), we compared the abundance of endogenous and exogenous Halo-tagged RARα and RXRα using flow cytometry (*Cattoglio et al., 2019*; *Figure 2—figure supplement 1A*). Average cellular abundance of endogenous RARα and RXRα obtained from biological replicates of each homozygous clone were similar (*Figure 2B*). In contrast, expression of exogenous RARα-Halo was approximately four times that of the endogenous protein, while expression of Halo-RXRα was nearly twenty times that of endogenous RXRα (*Figure 2B*). Chromatin binding of overexpressed RARα was reduced by approximately half compared to endogenous ($f_{bound}$ = 27 ± 0.72%) (*Figure 2C*, *Figure 2—figure supplement 1B*), while that of overexpressed RXRα was decreased even more dramatically to 11 ± 0.75% - a value barely above the $f_{bound}$ of the Halo-NLS control (*Figure 2C*, *Figure 2—figure supplement 1B*). Using nuclear fluorescence intensity as a rough proxy for protein concentration in individual cells, we observed a negative correlation in single cells between TF concentration and $f_{bound}$ (*Figure 2—figure supplement 1C*). These results imply that chromatin binding of both RARα and RXRα in U2OS cells is saturable - i.e., the total number of chromatin-bound molecules does not increase indefinitely with expression level but is in some way limited.

## RARα limits chromatin binding of RXRα

We next assessed how overexpression of RARα and RXRα affects $f_{bound}$ of the endogenous proteins by stable integration of SNAP-tagged RARα or RXRα transgenes in Halo-KI RARα and RXRα cell lines (*Figure 2D*). As controls, we also stably integrated transgenes expressing SNAP-NLS or a SNAP-tagged dimerization-incompetent RARα (RARα$^{RR}$) (*Bourguet et al., 2000*; *Zhu et al., 1999*; *Figure 2D*). We validated disruption of the RARα$^{RR}$-RXRα interaction using Rosetta modeling (*Shringari et al., 2020*) and co-IP (*Figure 2—figure supplement 2A*). Using flow cytometry, fluorescent gels, and western blots, we first assessed if transgene expression alters expression of endogenous Halo-tagged RARα and RXRα (*Figure 2E*, *Figure 2—figure supplement 2B–D*). While no drastic changes in the cellular abundance of KI Halo-tagged RARα or RXRα was observed in the presence of SNAP, SNAP-RXRα, or RARα$^{RR}$-SNAP proteins, the abundance of KI Halo-tagged RARα was approximately halved when RARα-SNAP was overexpressed (*Figure 2E*, *Figure 2—figure supplement 2B*). This is likely distinct from previously reported ligand-dependent RARα degradation (*Tsai et al., 2023*; *Zhu et al., 1999*) (see Discussion).

We then carried out a series of fSMT experiments to understand how $f_{bound}$ of the endogenous RARα or RXRα is altered when its binding partner is present in excess (*Figure 2—figure supplement 4A and B*). Surprisingly, we found that the $f_{bound}$ of endogenous RARα-Halo (47 ± 1%) was largely unchanged upon expression of RXRα-SNAP (49 ± 1%), consistent with the control SNAP (47 ± 1%) (*Figure 2F*, *Figure 2—figure supplement 4B*), implying that RARα $f_{bound}$ is not limited by the availability of RXR. In contrast, RARα-SNAP expression significantly decreased chromatin binding of endogenous RARα-Halo to 29 ± 5% (*Figure 2F*, *Figure 2—figure supplement 4B*). However, overexpression of mutant RARα$^{RR}$-SNAP did not change the $f_{bound}$ of endogenous RARα (43 ± 4%) (*Figure 2F*, *Figure 2—figure supplement 4B*), suggesting that heterodimerization with RXRα is required for this effect.

In the reciprocal experiment with endogenous Halo-RXRα, the initial $f_{bound}$ (35 ± 1%) was barely altered by overexpression of SNAP (38 ± 1%) or RARα$^{RR}$-SNAP (35 ± 1%). As expected, overexpression of SNAP-RXRα reduced the $f_{bound}$ of endogenous Halo-RXRα to 16 ± 2%. In contrast to RARα, however, overexpression of RARα-SNAP increased the $f_{bound}$ of endogenous Halo-RXRα to 56 ± 1%. This result suggests that endogenous RXRα chromatin binding is limited by the availability of RAR (*Figure 2F*, *Figure 2—figure supplement 4B*), and not due to limiting amounts of the universal dimer core partner RXR as would be expected based on current models in the literature.

While it may seem paradoxical that RAR is limiting for RXR binding, given the similar number of molecules of endogenous RARα and RXRα per cell (*Figure 2B*), this is likely due to the presence of other endogenous RXR paralogs (see *Appendix 2—table 1* and Discussion). It is also probable that some fraction of RXR binding to chromatin arises from other T2NRs that can produce chromatin binding-competent dimers with RXR (*Figure 1A*, *Appendix 2—table 1*, and Discussion). Notwithstanding these complexities, the results of the above SMT measurements in the presence of varying

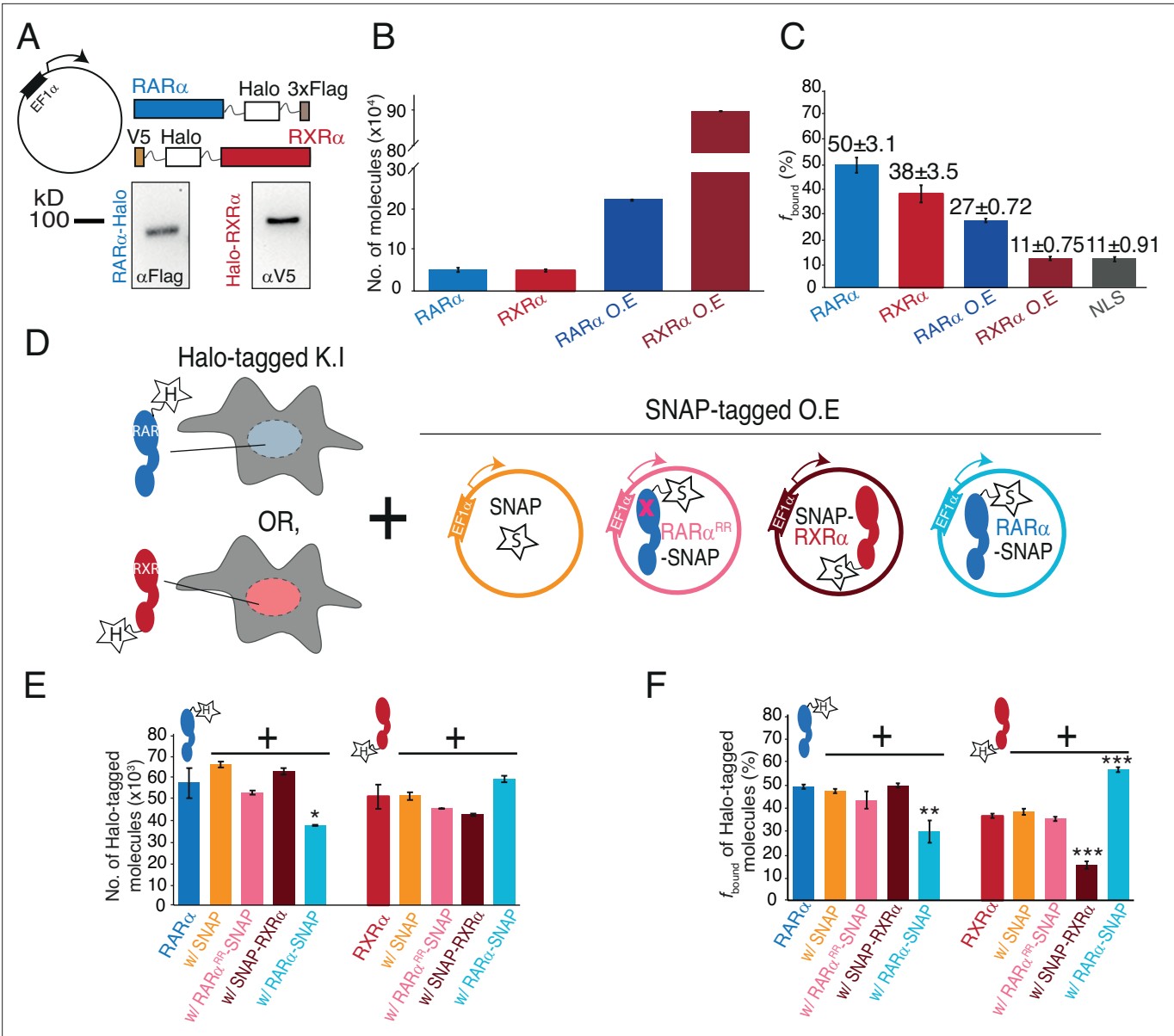

**Figure 2.** Chromatin binding of RARα and RXRα can be saturated and is limited by RARα. (**A**) Schematic and western blot of stably integrated EF1α promoter-driven Halo-tagged (HT) RARα (left) and RXRα (right) overexpression in wild-type (WT) U2OS cells. (**B**) Bar plot; y-axis shows number of HT knock-in (KI) and overexpressed (OE) RARα (blue) and RXRα (red) molecules quantified using flow cytometry. (**C**) Bar plot; y-axis depicts chromatin-bound fraction ($f_{bound}$%) of KI and OE RARα, RXRα compared to Halo-NLS (control). (**D**) Assay condition schematics to determine which of the partners in the RARα/RXRα heterodimer complex is limiting for chromatin binding; parental KI HT RARα or RXRα clones with OE SNAP (orange), SNAP-RXRα (brown), RARα-SNAP (light blue), and RARα$^{RR}$-SNAP (pink) using stably integrated EF1α promoter-driven transgene. (**E**) Bar chart; y-axis denotes number of KI HT RARα and RXRα molecules (depicted as blue and red cartoon respectively with 'H' labeled star attached) in presence or absence of transgene products. Error bars denote stdev of the mean from three biological replicates. (**F**) Bar plot showing $f_{bound}$% of KI HT RARα and RXRα in presence or absence of exogenously expressed SNAP proteins. Error bars for (**B**), (**E**) denote stdev of the mean from three biological replicates. Error bars for (**C**), (**F**) represent stdev of bootstrapping mean. p-Value≤0.001 (***), ≤0.01 (**), and ≤0.05 (*).

The online version of this article includes the following source data and figure supplement(s) for figure 2:

**Source data 1.** Includes one file (pdf) containing and explaining each of the raw images in *Figure 2A*, along with the loading controls.

**Source data 2.** Includes eight raw images (tif) for western blots displayed in *Figure 2A*, along with the loading controls.

**Figure supplement 1.** Cellular abundance and diffusion behavior of overexpressed Halo-tagged (HT) RARα and RXRα.

**Figure supplement 2.** Verification of mutant RARα (RARα$^{RR}$) and exogenous SNAP-tagged protein expression.

**Figure supplement 2—source data 1.** Includes one file (pdf) containing and explaining each of the raw images shown in *Figure 2—figure*

*Figure 2 continued on next page*

*Figure 2 continued*

supplement 2A.

**Figure supplement 2—source data 2.** Includes four raw images (tif) for co-immunoprecipitation (co-IP) western blots displayed in *Figure 2—figure supplement 2A*.

**Figure supplement 2—source data 3.** Includes one file (pdf) containing and explaining each of the raw fluorescent gel images shown in *Figure 2—figure supplement 2C*.

**Figure supplement 2—source data 4.** Includes four raw images (tif) for fluorescent gels displayed in *Figure 2—figure supplement 2C*.

**Figure supplement 2—source data 5.** Includes one file (pdf) containing and explaining each of the raw western blot images for RAR-Halo w/ SNAP transcripts, as shown in *Figure 2—figure supplement 2D*.

**Figure supplement 2—source data 6.** Includes one file (pdf) containing and explaining each of the raw western blot images for Halo-RXR w/ SNAP transcripts, as shown in *Figure 2—figure supplement 2D*.

**Figure supplement 2—source data 7.** Includes six raw images (tif) of western blot images for RAR-Halo w/ SNAP transcripts displayed in *Figure 2—figure supplement 2D*.

**Figure supplement 2—source data 8.** Includes six raw images (tif) of western blot images for Halo-RXR w/ SNAP transcripts displayed in *Figure 2—figure supplement 2D*.

**Figure supplement 3.** Live-cell confocal images to verify protein expression.

**Figure supplement 4.** Fast single-molecule tracking (fSMT) of Halo-tagged (HT) RARα and RXRα with (w/) different transgene products.

**Figure supplement 5.** Single-molecule tracking (SMT) analysis to verify change in $f_{bound}$% with respect to mean intensity.

amounts of partner TFs allow us to infer that endogenous RXRα is likely in excess while RXR partners are limiting in U2OS cells.

## PAPA-SMT shows that RARα-RXRα dimerization correlates with chromatin binding

The above results support a model in which the total RXR pool (including all paralogs) is in stoichiometric excess of its partners in U2OS cells. To more directly monitor RXR-RAR interactions, we employed a recently developed SMT assay by our lab, PAPA (*Graham et al., 2022*). PAPA detects protein-protein interactions by using excitation of a 'sender' fluorophore with green light to reactivate a nearby 'receiver' fluorophore from a dark state (*Graham et al., 2022*; *Figure 3A*). As an internal control, violet light is used to induce direct reactivation (DR) of receiver fluorophores independent of their proximity to the sender (*Figure 3A*). See Appendix 1 for a more detailed description of the steps involved in a PAPA-SMT experiment.

We applied PAPA-SMT to cell lines expressing SNAP-tagged RARα and RXRα transgenes in the background of Halo-tagged endogenous RXRα or RARα, respectively (*Figure 3B*). As controls, we also imaged SNAP-3×NLS and RARα^RR-SNAP, which are not expected to interact with Halo-RARα or RXRα. First, we plotted the number of single-molecule localizations reactivated by green and violet light in each cell (*Figure 3B*). As expected, a low level of background reactivation by green light was observed for SNAP negative control, reflecting the baseline probability that an unbound SNAP molecule will at some low level be close enough to Halo to observe PAPA (*Figure 3B*, orange points) (*Graham et al., 2022*). Violet light-induced DR and green light-induced PAPA were linearly correlated as expected, since both are proportional to the number of receiver molecules. However, a greater PAPA signal was seen for the Halo-RXRα → RARα-SNAP and RARα-Halo → SNAP-RXRα combinations than for the negative control, consistent with direct protein-protein interactions (*Figure 3B*). The relation between DR and PAPA was sublinear (*Figure 3B*, left and middle panel, see residuals of the linear fit in *Figure 3—figure supplement 1B*), consistent with saturation of binding to the Halo-tagged component. In contrast, the ratio of PAPA to DR for Halo-RXRα → RARα^RR-SNAP was similar to the SNAP negative control, confirming that PAPA signal depends on a functional RAR-RXR interaction interface (*Figure 3B*, right panel).

Next, we compared the diffusion spectra of molecules reactivated by DR and PAPA (*Figure 3C*). For both Halo-RXRα → RARα-SNAP and RARα-Halo → SNAP-RXRα combinations, PAPA trajectories had a substantially higher chromatin-bound fraction than DR trajectories, indicating that a greater proportion of SNAP-tagged RARα/RXRα binds chromatin when it is in complex with its Halo-tagged partner. Only a slight shift in bound fraction was seen for RARα^RR-SNAP ($f_{bound,DR}$ = 7.5 ± 0.6%; $f_{bound,PAPA}$

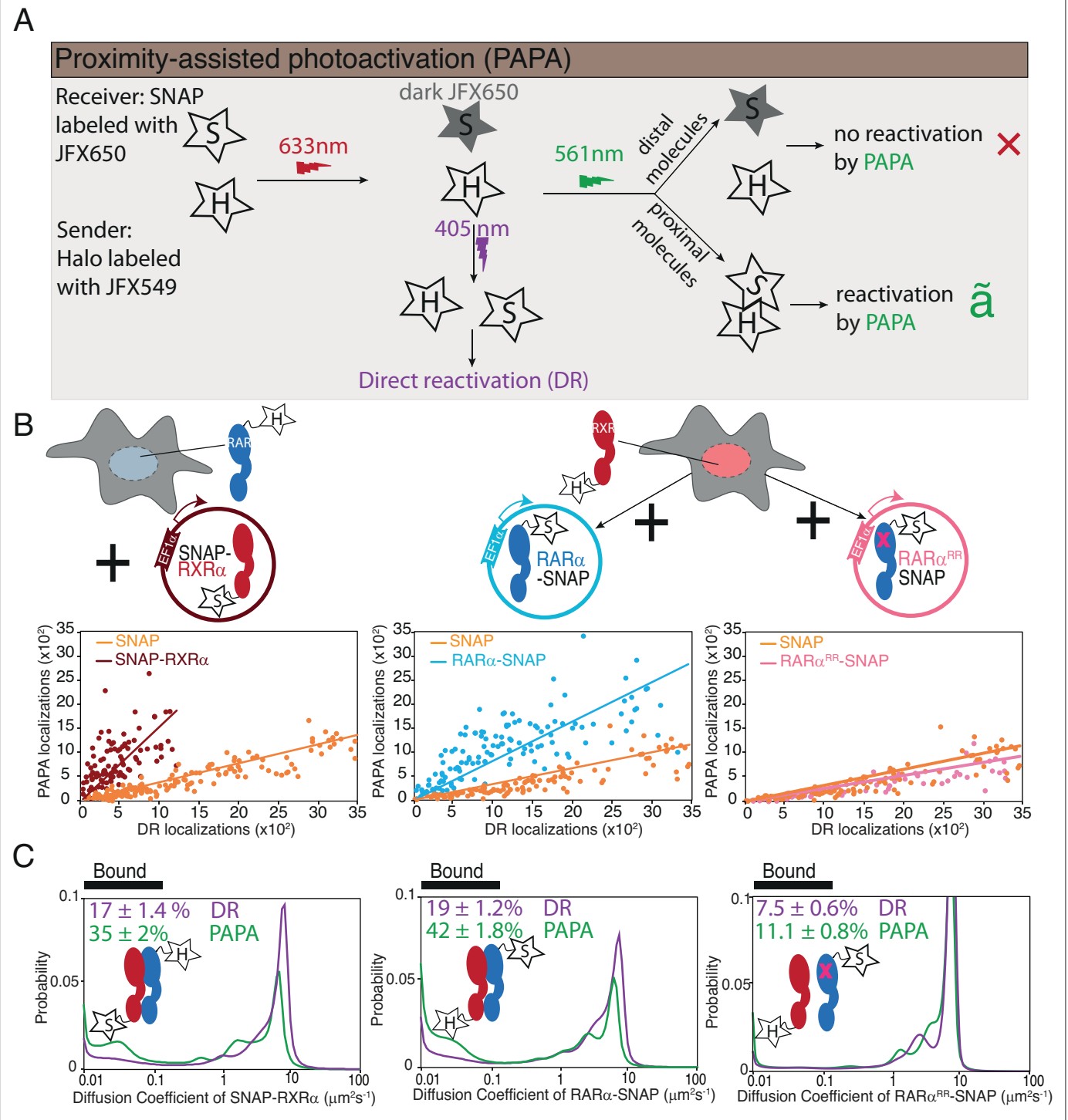

**Figure 3.** Proximity-assisted photoactivation (PAPA)-single-molecule tracking (SMT) shows direct interaction between Halo-tagged (HT) knock-in (KI) and SNAP-tagged overexpressed RARα and RXRα in live cells. (**A**) Schematic illustrates how PAPA signal is achieved. First, SNAP-tagged (ST) protein is labeled with 'receiver' fluorophore like JFX650 (star with letter 'S') and HT protein is labeled with 'sender' fluorophore like JFX549 (star with letter 'H'). When activated by intense red light the receiver fluorophore goes into a dark state (gray star with S). Upon illumination by green light, the receiver and sender molecules distal to one another do not get photoactivated (red X) but receiver SNAP molecules proximal to the sender gets photoactivated (green ✓). Pulses of violet light can induce direct reactivation (DR) of receiver independent of interaction with the sender. PAPA experiments. (**B**) Plots showing PAPA versus DR reactivation. (**C**) Diffusion spectra of PAPA and DR trajectories obtained for ST proteins for the represented conditions; parental HT RARα KI cell, stably expressing RXRα-SNAP (brown, left panel), as well as parental HT RXRα KI cell stably expressing RARα-SNAP (light blue, middle panel) and RARα^RR-SNAP (light pink, right panel). A linear increase in PAPA versus DR reactivation is seen for non-interacting SNAP controls

*Figure 3 continued*

and a sublinear increase is seen for interacting SNAP proteins. Respective colored lines show linear fits of the data (see residuals in ***Figure 3—figure supplement 1B***). SNAP control data are re-plotted in middle and right panels. Cartoon inside diffusion spectra depicts if the expressed HT and ST proteins are expected to interact or not. $f_{bound}$% errors represent stdev of bootstrapping mean.

The online version of this article includes the following figure supplement(s) for figure 3:

**Figure supplement 1.** Controls for proximity-assisted photoactivation (PAPA) experiments.

$= 11.1 \pm 0.8$%), comparable to that seen for the SNAP negative control ($f_{bound,DR} = 7.1 \pm 0.5$% and $f_{bound,PAPA} = 11.2 \pm 0.9$% for Halo-RXRα; $f_{bound,DR} = 7.3 \pm 0.4$% and $f_{bound,PAPA} = 9.3\pm0.7$ for RARα-Halo) (see Discussion), confirming that RARα$^{RR}$ fails to form chromatin binding-competent heterodimers with RXR.

The enrichment of chromatin-bound molecules in the PAPA-reactivated population of both RAR and RXR is consistent with heterodimerization mediating chromatin binding, that is further validated by the low $f_{bound}$ of the dimerization-incompetent mutant.

## Discussion

According to the current consensus models for how nuclear receptor interaction networks operate, specific T2NRs members compete for a limited pool of the 'core' partner RXR in cells, thus establishing a competitive regulatory network (***Chan and Wells, 2009***; ***Fadel et al., 2020***; ***Rehó et al., 2023***; ***Wang et al., 2005***; ***Wood, 2008***; ***Yoshikawa et al., 2003***). However, previous in vivo studies were carried out with overexpressed proteins, limiting their ability to accurately address this model of competition (***Fadel et al., 2020***; ***Rehó et al., 2023***). Here, we used SMT in live cells with endogenously tagged proteins and carefully controlled protein levels of two players, RARα and RXRα, in the T2NRs dimerization network, thereby directly addressing a fundamental question - whether RAR (partner) or RXR (core) is limiting for chromatin association.

In stark contrast to the generally accepted T2NR competition model, our results reveal that in U2OS cells, formation and chromatin binding of RAR-RXR heterodimers are limited by the concentration of RAR and not the core subunit RXR (***Figure 4***). PAPA-SMT directly confirmed in vivo that the association with RXRα promotes chromatin binding of RARα and vice versa (***Figure 3C***). However, unexpectedly, overexpression of RARα increases $f_{bound}$ of endogenous RXRα, while the reverse is not true (***Figure 2F***), indicating that RXR is not limiting but is rather constrained by the availability of its binding partners. Note that even though the number of RXRα and RARα molecules is about the same and the heterodimer complex has a predicted 1:1 stoichiometry (***Rastinejad, 2022***; ***Rastinejad et al., 2000***), the $f_{bound}$ of endogenous RARα (50%) and RXRα (38%) differed by about 10% (***Figures 1D and 2B***). This surprising result could be explained by expression of additional RXR isoforms (most likely RXRβ) in U2OS cells (***Figure 1A***, ***Appendix 2—table 1***, ***Figure 1—figure supplement 2B***). The fact that the $f_{bound}$ of overexpressed RARα decreases only twofold, even though it is in fourfold excess over endogenous RXRα, likewise suggests that there are other heterodimerization partners of RARα available (***Figure 2A and B***, see ***Appendix 2—table 1***). Moreover, the $f_{bound}$ of overexpressed RXR is essentially the same as the NLS control, indicating this excess core partner remains nearly totally unbound (***Figure 2C***). This is consistent with endogenous RXR already being in excess such that any additional RXR would remain mostly monomeric and unbound to chromatin. Hence, despite not directly measuring the protein levels of all RXR isoforms, we can still deduce in U2OS cells that RXR is in excess relative to its binding partners.

Control of RAR-RXR heterodimer concentration by RAR abundance makes sense considering the observation that RAR protein levels appear to be regulated by multiple feedback mechanisms: First, we find that overexpression of RARα significantly lowers the expression of endogenous RARα (***Figure 2E***) implying either that RAR participates in negative autoregulation at the transcriptional level or, that an excess of RAR causes instability at the protein level. We favor the latter possibility because overexpression of RARα reduces not only the expression of endogenous RARα but also its $f_{bound}$ (***Figure 2F***), indicating a reduction in dimerization and chromatin binding. It is possible that endogenous RAR may be more readily degraded when not bound by RXR, analogous to what was recently shown for the c-MYC/MAX heterodimers (***Mark et al., 2023***). Second, as has been previously reported, RARα expression decreases upon ligand treatment (***Ismail and Nawaz, 2005***; ***Kopf et al.,***

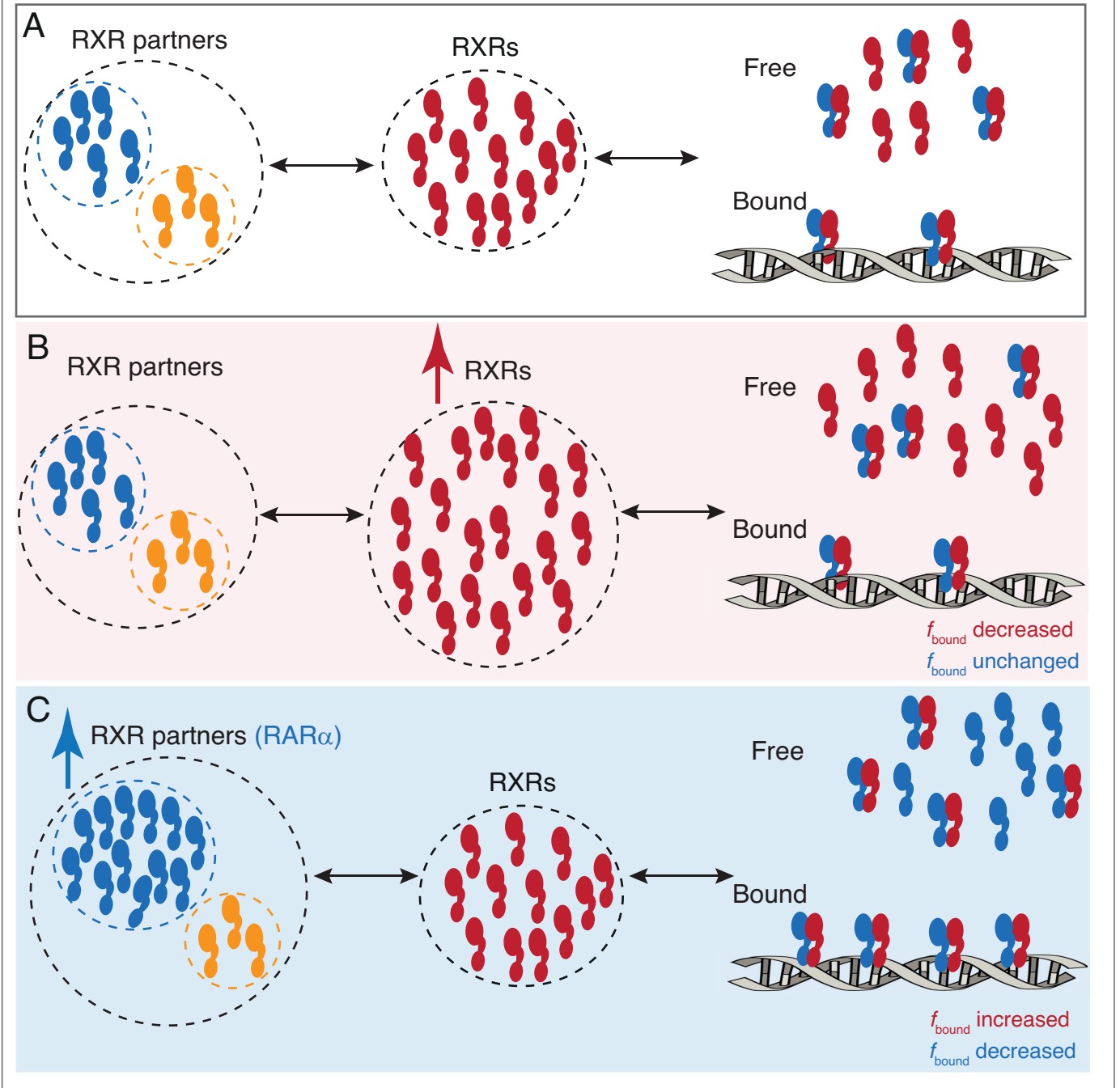

**Figure 4.** A model for RARa limited chromatin binding of RARa-RXRa heterodimers. (**A**) Pool of RXRα (red) and RXR partners (RARα - blue, other Type II nuclear receptors [T2NRs] - yellow) along with some number of chromatin-bound RARα-RXRα heterodimers exist under normal conditions. (**B**) When the pool of free RXRα is increased, the number of chromatin-bound RARα-RXRα heterodimers does not change. (**C**) When the pool of RARα is increased, chromatin binding of RARα-RXRα heterodimers increases, until it reaches saturation. Note: For simplicity we have omitted to show heterodimerization of other T2NRs (yellow) with RXRα (red).

*2000*; *Osburn et al., 2001*; *Tsai et al., 2023*; *Zhu et al., 1999*). Curiously, a 50% reduction in RARα upon addition of ligand did not affect the $f_{bound}$ of either RARα or RXRα (*Figure 1—figure supplement 3A and B*), suggesting that most chromatin binding of endogenous RXRα in U2OS cells depends on heterodimerization partners other than RARα (see *Appendix 2—table 1*).

In contrast to the generally accepted model, we thus envision a network of T2NR heterodimers not always driven by competition for the core TF partner (RXR). Instead, an excess of RXR may ensure independent regulation of the different T2NRs without disrupting crosstalk between them (*Figure 4*).

In this first study, we have not examined other T2NRs expressed in U2OS cells or whether there is a similar excess of RXR in other cell types. Therefore, we cannot rule out that competition for RXR between T2NRs occurs in some cases since different cell types express RXR and partner T2NRs in varying amounts. It seems likely that chromatin binding by any given T2NR will depend on the concentrations of all other T2NRs. It is thus important to determine how interactions within the dimerization network are perturbed upon up- or down-regulation of one or more T2NRs, especially since dysregulation of T2NR expression has been reported in several types of cancers as well as other diseases (*Brabender et al., 2005*; *Collins-Racie et al., 2009*; *Frigo et al., 2021*; *Long and Campbell, 2015*). Here, we have shown that SMT and PAPA-SMT provide one empirical way to determine which set of components in a dimerization network is stoichiometrically limiting in a given cell type, without having to measure the concentration of every T2NR or the affinity of every interaction. The basic framework we have established here could also be extended to probe the effect of ligands or small molecules on the T2NR interaction network or other dimerization networks seen in bHLH or leucine zipper family of TFs, providing useful information about critical regulatory network interactions often implicated in diseases.

## Methods

### Cell culture and stable cell line generation

U2OS cells were grown in Dulbecco's modified Eagle's medium (DMEM) with 4.5 g/l glucose supplemented with 10% fetal bovine serum (FBS) (HyClone, Logan UT, Cat. #AE28209315), 1 mM sodium pyruvate (Thermo Fisher 11360070), L-glutamine (Sigma #G3126-100G), GlutaMAX (Thermo Fisher #35050061), and 100 U/ml penicillin-streptomycin (Thermo Fisher #15140122) at 37°C and 5% $CO_2$. Cells were subcultured at a ratio of 1:4 to 1:10 every 2–4 days for no longer than 30 passages. Regular mycoplasma testing was performed using polymerase chain reaction (PCR). Phenol containing media was used for regular cell culture and phenol red-free DMEM (Thermo Fisher #21063029) supplemented with 10% FBS and 100 U/ml penicillin-streptomycin was used for imaging.

Stable cell lines expressing the exogenous gene products (*Appendix 2—table 2*) were generated by PiggyBac transposition and antibiotic selection. Gibson assembly was used to clone genes of interest into a PiggyBac vector containing a puromycin or neomycin resistant gene. Plasmids were purified by Zymo midiprep kit (Zymo D4200) and all cloning was confirmed by Sanger sequencing. Cells were transfected by nucleofection using the Lonza Cell line Nucleofector Kit V (Lonza, Basel, Switzerland, #VVCA1003) and the Amaxa Nucleofector II device. For each transfection cells were plated 1–2 days before nucleofection in a six-well plate until they reach 70–90% confluency. Cells were trypsinized, resuspended in DMEM media, and centrifuged at 200×*g* for 2 min before the media was aspirated. Cells were then resuspended in 100 μl Lonza transfection reagent (82 μl Kit V solution+16 μl of Supplement #VVCA1003) containing 0.4 μg of SuperPiggyBac transposon vector and 0.8 μg of donor PiggyBac plasmid and transferred to an electroporation cuvette. Cells were electroporated using program X-001 on the Amaxa Nucleofector II (Lonza). Transfected cells were cultured in DMEM growth media without any antibiotics for 24–48 hr and then selected for 10 days with 1 μg/ml puromycin (Thermo Fisher #A1113803) or 1 mg/ml neomycin (G418 Sulfate, Thermo Fisher #10131027). After selection polyclonal cell lines were maintained in the selection media containing required antibiotics.

For ligand treatment, 100 μM atRA stock was prepared by dissolving atRA powder (CAS No: 302794, Sigma-Aldrich #R2625) in dimethyl sulfoxide (DMSO) (Sigma-Aldrich #D2650) and was diluted 1:100,000 or 1:1000 in growth media to final concentration of 1 nM or 100 nM respectively. The same volume of DMSO used for 100 nM atRA treatment (0.1%) is used for the control group as a condition without atRA treatment. Cells were treated 24 hr in either atRA or DMSO alone before imaging.

### Genome editing cell lines

KI cell lines were generated as previously described (*Hansen et al., 2017*) with some modifications. Halo-tagging of endogenous RARα was described in our previously published work (*Heckert et al., 2022b*) which we have further validated for the current study. For Halo-tagging we designed sgRNAs using CRISPOR web tool (*Concordet and Haeussler, 2018*). Since exon 1 of RXRα is very short (only 9 aa) we chose to gene edit at the start of exon 2 for a successful tagging. sgRNAs were cloned into

the Cas9 plasmid (a gift from Frank Xie) under the U6 promoter (Zhang Lab) with an mVenus reporter gene under the PGK promoter. Repair vectors were cloned in a basic pUC57 backbone for N-terminal tagging and pBluescript II SK (+) (pBSKII+) backbone for C-terminal tagging, with 500 bp left and right homology arms on either side of the Halo-tag sequence. Two guide/repair for N-terminal and three guide/repair vector pairs for C-terminal were attempted; only-N-terminal clones were ultimately recovered. Each sgRNA/donor pair were transfected to approximately 1 million early passage U2OS cells, at 1:3 ratio of sgRNA/donor (total 5 µg DNA) and plated in a six-well plate. 48 hr after transfection, Venus-positive cells were FACS sorted and cultured for another 7–10 days. Then, Halo-positive cells (stained with TMR) were sorted individually into single wells of 96-well plates and cultured for another 12–14 days. Clones were expanded and genotyped using PCR. PCR was done using one primer upstream of the left homologous arm and the other primer downstream of the right homologous arm. Another PCR with either external primer paired with a corresponding internal primer located in the Halo-tag coding region was done for further validation. Homozygous clones with the correct genotype (V5-Halo-RXRα clones C10, D6, and D9) were confirmed by Sanger sequencing and western blotting.

## Antibodies

The following antibodies were used for western blotting: mouse monoclonal anti-RARα [H1920] (Abcam, #ab41934) diluted at 1:400, rabbit monoclonal anti-RXRα [EPR7106] (Abcam, #ab125001) diluted at 1:500, rabbit polyclonal anti-RXRα (Proteintech, #212181AP), mouse monoclonal anti-V5 tag (Thermo Fisher, #R960-25) diluted at 1:5000, mouse monoclonal anti-FLAG [M2] (Sigma-Aldrich, #F1804) diluted at 1:5000, mouse monoclonal anti-Halo (Promega, #G9211) diluted at 1:500, mouse monoclonal anti-TBP (Abcam, #ab51841) diluted at 1:5000, rabbit polyclonal anti-Centrin 2 (Proteintech, # 158771AP) diluted at 1:1000, goat polyclonal anti-mouse IgG light chain specific (Jackson ImmunoResearch, # 115035174) diluted at 1:10,000, mouse monoclonal anti-rabbit IgG light chain specific (Jackson ImmunoResearch, #211032171) diluted at 1:10,000.

The following antibodies were used for co-IP: rabbit polyclonal anti-V5 (Abcam, # ab9116) for immunoprecipitation, mouse monoclonal anti-FLAG [M2] (Sigma-Aldrich, #F1804) diluted at 1:5000 and mouse monoclonal anti-V5 tag (Thermo Fisher, #R960-25) diluted at 1:5000 for blotting.

## Western blotting

For western blots, cells growing in six-well or 10 cm plates at 80–90% confluency were scraped and pelleted in ice-cold phosphate-buffered saline (PBS) containing protease inhibitors. Cell pellets were resuspended using 500–1000 µl hypotonic buffer (100 mM NaCl, 25 mM HEPES, 1 mM MgCl$_2$, 0.2 mM EDTA, 0.5% NP-40 alternative) containing protease inhibitors - 1× aprotinin (Sigma, #A6279, diluted 1:1000), 1 mM benzamidine (Sigma, #B6506), 0.25 mM PMSF (Sigma #11359061001), and 1× cOmplete EDTA-free Protease Inhibitor Cocktail (Sigma, #5056489001) along with 125 U/ml of benzonase (Novagen, EMD Millipore #71205-3). Resuspended cells were gently rocked at 4°C for at least 2 hr after which 5 M NaCl was added. Cells were left rocking for extra 30 min at 4°C and then centrifuged at max speed at 4°C. Supernatants were quantified using Bradford and 15–20 µg was loaded on 8% Bis-Tris SDS gel. Wet transfer to 0.45 µm nitrocellulose membrane (Thermo Fisher #45004031) was performed in a transfer buffer (15 mM Tris-HCl, 20 mM glycine, 20% methanol) for 60–80 min at 100 V, 4°C. Membranes were blocked in 10% non-fat milk in 0.1% TBS-Tween (TBS-T) for 1 hr at room temperature (RT) with agitation. Membranes were then blotted overnight with shaking at 4°C with primary antibodies diluted in TBS-T with 5% non-fat milk. After 5×5 min washes in 0.1% TBS-T, membranes were incubated at RT with HRP-conjugated light chain secondary antibodies diluted 1:10,000 in TBS-T with 5% non-fat milk, for 1 hr with agitation. After 5×5 min washes in 0.1% TBS-T, membranes were incubated for 2 min in freshly prepared Perkin Elmer LLC Western Lightning Plus-ECL, enhanced Chemiluminescence Substrate (Thermo Fisher, #509049326). Finally, membranes were imaged with Bio-Rad Chemidoc Imaging System (Bio-Rad, Model No: Universal Hood III).

## Luciferase assays

pGL3-RARE luciferase, a reporter containing firefly luciferase driven by SV40 promoter with three RAREs, a gift from T Michael Underhill (Addgene plasmid 13456; addgene:13458; RRID:Addgene_13458; *Hoffman et al., 2006*) was used for luciferase assays. A pRL-SV40 vector (Promega

#E2261) expressing Renilla luciferase with SV40 promoter was used as a control to normalize luciferase activity. Cells were plated on six-well plate at least 48 hr before being co-transfected with 200 ng pGL3-RARE-luciferase and 10 ng Renilla luciferase vector, using TransIT-2020 Transfection Reagent (Mirus Bio, #MIR5404). A day after transfection, cells were treated with 100 nM atRA or 0.1% DMSO (control). Luciferase assay was performed the next day, using Dual-Luciferase Reporter Assay System (Promega, #E1910) according to the manufacturer's protocol, on the Glomax Luminometer (Promega). The relative luciferase activity was calculated by normalizing firefly luciferase activity to the Renilla luciferase activity to control for transfection efficiency.

## Co-immunoprecipitation

For co-IP experiments, Cos7 cells were plated to 2×15 cm dishes for each condition, at 60–70% confluency. DNA Lipofectamine 3000 (Thermo Fisher #L3000015) was used to co-transfect the cells with plasmids expressing RARα-Halo-3×FLAG and RXRα-V5 or V5-Halo-RXRα and RARα-3×FLAG or RARα$^{RR}$-Halo-3×FLAG and RXRα-V5. According to the manufacturer's instructions, 500 µl Opti-MEM medium (Thermo Fisher #31985062) was combined with 20 µl P3000 reagent and the plasmids for each condition. To control for the ratio of transfected DNA to the reagents, total transfected DNA mass was kept at 10 µg for each condition using empty pBSK vector (Addgene # 212205). 500 µl Opti-MEM medium containing 20 µl Lipofectamine 3000 reagent was subsequently added to the plasmid containing mixture and incubated at RT for 15 min, after brief pipetting to mix the solutions. The mixture was then divided equally to the cells plated in 2×15 cm dishes. COS7 cells were cultured in DMEM media with 4.5 g/l glucose supplemented with 10% FBS, 1 mM sodium pyruvate, L-glutamine, GlutaMAX, and 100 U/ml penicillin-streptomycin and maintained at 37°C and 5% $CO_2$.

Cells were collected from plates 48 hr after transfection by scraping in ice-cold 1× PBS with protease inhibitors (1× aprotinin, 0.25 mM PMSF, 1 mM benzamidine, and 1× cOmplete EDTA-free Protease Inhibitor Cocktail). Collected cells were pelleted, flash-frozen in liquid nitrogen, and stored at –80°C. On the day of co-IP experiments, cell pellets were thawed on ice, resuspended to 700 µl of cell lysis buffer (10 mM HEPES pH 7.9, 10 mM KCl, 3 mM $MgCl_2$, 340 mM sucrose [1.16gr], and 10% glycerol) with freshly added 10% Triton X-100. Cells were rocked at 4°C for 8 min to allow lysis and centrifuged for 3 min at 3000×g. The cytoplasmic fraction was removed, and the nuclear pellets were resuspended in 1 ml hypotonic buffer (100 mM NaCl, 25 mM HEPES, 1 mM $MgCl_2$, 0.2 mM EDTA, 0.5% NP-40 alternative) containing protease inhibitors and 1 µl benzonase. After rocking for 2–3 hr at 4°C, the salt concentration was adjusted to 0.2 M NaCl final and the lysates were rocked for another 30 min at 4°C. Supernatant was removed after centrifugation at maximum speed at 4°C for 20 min and quantified by Bradford. Typically, 1 mg of protein was diluted in 1 ml 0.2 mM co-IP buffer (200 mM NaCl, 25 mM HEPES, 1 mM $MgCl_2$, 0.2 mM EDTA, 0.5% NP-40 alternative) with protease inhibitors and cleared for 2 hr at 4°C with magnetic Protein G Dynabeads (Thermo Fisher, #10009D) before overnight immunoprecipitation with 1 µg per 250 µg of proteins of either normal serum IgGs or specific antibodies as listed above. Some precleared lysates were kept overnight at 4°C as input. Magnetic Protein G Dynabeads were also precleared overnight with 0.5% BSA at 4°C. Next day the precleared Protein G dynabeads were added to the antibody containing samples and incubated at 4°C for 2 hr. Samples were briefly spun down and placed in a magnetic rack at 4°C for 5 min to remove the co-IP supernatant. After extensive washes with the co-IP buffer, the proteins were eluted from the beads by boiling for 10 min in 1× SDS-loading buffer and analyzed by SDS-PAGE and western blot.

## Flow cytometry

Cells were grown in six-well dishes. On the day of the experiment, cells were labeled with 500 nM Halo-TMR for 30 min, followed by one quick wash with 1× PBS and 15 min wash in dye-free DMEM before trypsinization. Cells were pelleted after centrifugation, resuspended in fresh medium, filtered through 40 µm filtration unit, and placed on ice until fluorescence read out by Flow Cytometry (within 30 min). Using an LSR Fortessa (BD Biosciences) flow cytometer, live cells were gated using forward and side scattering and TMR fluorescence emission readout was filtered using 610/20 band pass filter after excitation with 561 nm laser. Mean fluorescence intensity of the samples and absolute abundance were calculated as described in *Cattoglio et al., 2019*, using a previously quantified Halo-CTCF cell line as a standard (*Cattoglio et al., 2019*).

## Cell preparation and dye labeling for imaging

For confocal imaging ~50,000 cells were plated in tissue culture-treated 96-well microplate (Perkin Elmer, #6055300) a day before imaging. Cells were labeled with Halo and/SNAP for 1 hr and incubated in dye-free media for 20 min after a quick 1× PBS wash to remove any free dye. Phenol-free media containing Hoechst (1 µM) was added to the cells and incubated for 1 hr before proceeding with imaging.

For SMT, 25 mm circular No. 1.5H precision coverglass (Marienfield, Germany, 0117650) were sonicated in ethanol for 10 min, plasma cleaned then stored in 100% isopropanol until use. ~250,000 U2OS cells were plated on sonicated, and plasma cleaned coverglass placed in six-well plates. For ligand treatment, cells were treated with DMSO (control, 0.1%), 1 nM atRA or 100 nM atRA, a day after plating cells in the coverslip and a day before imaging. After ~24–48 hr, cells were incubated with 100 nM PA-JFX549 dye in regular culture media for 30 min, followed by 4×30 min incubations with dye-free culture media at 37°C. A quick 2× PBS wash was interspersed between each 30 min dye-free media incubation. After the final wash, coverslips with plated cells facing upward were transferred to Attofluor Cell Chambers (Thermo Fisher, # A7816), and phenol-free media added for imaging. For Halo-tagged RARα and RXRα homozygous clones that were stably integrated with SNAP-tagged RARα, RXRα, or control transgenes under EF1α promoter; cells were double-labeled with 100 nM HaloTag ligand (HTL) PA-JFX549 and SNAP tag ligand (STL) 50 nM SF-650 simultaneously. The 30 min labeling step was followed by the same wash steps to remove free dye as described above, before transferring the coverslips to Attofluor Cell Chambers and imaging.

For PAPA-SMT experiments, polyclonal U2OS cells stably expressing SNAP-tagged RARα or RXRα under EF1α promoter within the endogenous Halo-tagged RARα and RXRα homozygous clones were used. ~400,000 polyclonal cells were plated in glass-bottom dishes (MatTEK P35G-1.5–20-C) and stained overnight with 50 nM JFX549 HTL and 5 nM JFX650 STL in phenol red-free DMEM (Thermo Fisher #21063029). Next day, cells were briefly rinsed twice with 1× PBS, incubated twice for 30 min in phenol red-free DMEM to remove free dye, and exchanged into fresh phenol red-free medium more before imaging.

## Confocal imaging

For confocal imaging, endogenously Halo-tagged cells were labeled with Halo ligand JFX549 (100 nM) and Hoechst (1.6 µM); endogenously Halo-tagged cells overexpressing SNAP-tagged proteins were labeled with Halo ligand JFX549 (100 nM), SNAP ligand SF650 (25 nM), and Hoechst (1.6 µM). Imaging was performed at the UC Berkeley High Throughput Screening facility on a Perkin Elmer Opera Phenix equipped with 37°C and 5% $CO_2$, using a built-in ×40 water immersion objective.

## Live-cell SMT

All SMT experiments were performed using a custom-built microscope as previously described (*Hansen et al., 2017*). Briefly, a Nikon TI microscope was equipped with a ×100/NA 1.49 oil immersion total internal reflection fluorescence (TIRF) objective (Nikon apochromat CFI Apo TIRF 100× Oil), a motorized mirror, a perfect Focus system, an EM-CCD camera (Andor iXon Ultra 897), laser launch with 405 nm (140 mW, OBIS, Coherent), 488 nm, 561 nm, and 639 nm (1W, Genesis Coherent) laser lines, an incubation chamber maintaining a humidified atmosphere with 5% $CO_2$ at 37°C. All microscope, camera, and hardware components were controlled through the NIS-Elements software (Nikon).

For tracking endogenous as well as overexpressed Halo-tagged proteins, PA-JFX549 labeled cells were excited with 561 nm laser at 2.3 kW/cm² with single band-pass emission filter (Semrock 593/40 nm for PA-JFX549). All imaging was performed using highly inclined optical sheet (HILO) illumination (*Tokunaga et al., 2008*). Low laser power (2–3%) was used to locate and focus cell nuclei. Region of interest (ROI) of random size but with maximum possible area was selected to fit into the interior of the nuclei. Before tracking, 100 frames of continuous illumination with 549 laser (laser power 5%) at 80 ms per frame was recorded to later calculate mean intensity of each nuclei. After partial pre-bleaching (only for overexpressed Halo-tagged proteins), movies were taken with 1 ms pulses of full power of 561 nm illumination at the beginning of frame interval with camera exposure of 7 ms/frame, while the 405 nm laser (67 W/cm²) was pulsed during ~447 µs camera transition time.

Total of 30,000 frames were collected. 405 nm intensity was manually tuned to maintain low density (1–5 molecules fluorescent particles per frame).

For tracking experiments with endogenous Halo-tagged proteins in the presence of transgenic SNAP protein products, cells were dual-labeled with both PA-JFX549 and STL-SF650. Before tracking cells were excited with a 639 nm laser at 88 W/cm$^2$ and single band-pass emission filter set to Semrock 676/37 nm. Low laser power (2–3%) was used to locate and focus cell nuclei. As before, ROI of random size but with maximum possible area was selected to fit into the interior of the nuclei. The emission filter was switched to Semrock 593/40 nm band-pass filter for PA-JFX549, keeping the TIRF angle, stage XYZ position, and ROI the same. Cells were then excited with 549 nm and single molecules were tracked for 30,000 frames as described above.

Note, we estimated that to calculate $f_{bound}$ of RARα and RXRα with minimal bias and variation, we need to analyze at least 10,000 pooled trajectories from n≥20 cells (*Figure 1—figure supplement 1E*). Therefore, all analyses of fSMT to calculate $f_{bound}$ or $f_{bound}$ % presented in this paper are done accordingly. At least 8–10 movies were collected for each sample as one technical replicate on a given day. Two technical replicates on 2 separate days were collected to produce the reported results. As observed previously for SMT experiments (*McSwiggen et al., 2024*), the variance within collected data was largely due to cell-to-cell variance (*Figure 1—figure supplement 1E*).

## PAPA-SMT

PAPA-SMT imaging was performed using the same custom-built microscope described above using an automated system (*Graham et al., 2024*; *Walther et al., 2024*). Briefly, the microscope described above was programmed using Python and NIS Elements macro language code to raster in a grid over the coverslip surface, acquire an image in the JFX650 channel, identify cell nuclei, reposition the stage to center it on a target nucleus, resize the imaging ROI to fit the nucleus, pre-bleach JFX650 with red light (639 nm) for 5 s, and finally perform a PAPA-SMT illumination sequence. The illumination sequence consisted of 5 cycles of the following, at a frame rate of 7.48 ms/frame:

1. Bleaching: 200 frames of red light (639 nm), not recorded
2. Imaging: 30 frames of red light, one 2 ms stroboscopic pulse per frame, recorded
3. PAPA: 5 frames of green light (561 nm), not recorded
4. Imaging: 30 frames of red light, one 2 ms stroboscopic pulse per frame, recorded
5. Bleaching: 200 frames of red light, not recorded
6. Imaging: 30 frames of red light, one 2 ms stroboscopic pulse per frame, recorded
7. DR: 1 frames of violet light (405 nm), one 1 ms stroboscopic pulse, not recorded
8. Imaging: 30 frames of red light, one 2 ms stroboscopic pulse per frame, recorded

Laser power densities were approximately 67 W/cm$^2$ for 405 nm, 190 W/cm$^2$ for 561 nm, and 2.3 kW/cm$^2$ for 639 nm. This illumination sequence includes non-stroboscopic illumination periods to bleach fluorophores or place them back in the dark state (steps 1 and 5), and it records only the frames immediately before and after the PAPA and DR pulses. This is done to decrease file sizes and speed up subsequent analysis compared to our previous protocol (*Graham et al., 2022*).

## SMT and SMT-PAPA data processing and analysis

All SMT movies were processed using the open-source software package quot (*Heckert and Fan, 2022*). For SMT dataset quot package was run on each collected SMT movie using the following settings: [filter] start = 0; method = 'identity'; chunk_size = 100; [detect] method = 'llr'; k=1.0; w=15, t=18; [localize] method = 'ls_int_gaussian', window size = 9; sigma = 1.2; ridge = 0.0001; max_iter = 20; damp = 0.3; camera_gain = 109.0; camera_bg = 470.0; [track] method = 'euclidean'; pixel_size_μm=0.160; frame interval = 0.00748; search radius = 1; max_blinks = 0; min_IO = 0; scale - 7.0. The first 500 or 1000 frames of each movie were removed due to high localization density. To confirm that all movies were sufficiently sparse to avoid misconnections, a maximum number of 5 localizations per frame was maintained (although most frames had 1 or less). To infer the distribution of diffusion coefficients from experimentally observed trajectories, we used the state array method, which is publicly available at https://github.com/alecheckert/saspt (*Heckert et al., 2022a*; *Heckert et al., 2022b*). The following settings were used: likelihood_type = 'rbme'; pixel_size_μm=0.16; frame_interval = 0.00747; focal depth = 0.7; start_frame = 500 or 1000 frames. RBME likelihood for individual cells and occupations as the mean of the posterior distribution over state occupations marginalized on

diffusion coefficient is reported. For SMT movies in *Figure 2—figure supplement 1C*, mean intensity of each cell was calculated using the mean gray value tool in FIJI. Mean gray value of each frame (total 100 frames) of the pre-bleaching 80 ms movie was measured. Average mean gray value over the 100 frames is reported as mean intensity for each cell.

For PAPA-SMT quot package was run on each collected SMT movie using the following settings: [filter] start = 0; method = 'identity'; chunk_size = 100; [detect] method = 'llr'; k=1.2; w=15, t=18; [localize] method = 'ls_int_gaussian', window size = 9; sigma = 1.2; ridge = 0.0001; max_iter = 10; damp = 0.3; camera_gain = 109.0; camera_bg = 470.0; [track] method = 'conservative'; pixel_size_µm=0.160; frame interval = 0.00748; search radius = 1; max_blinks = 0; min_IO = 0; scale - 7.0. Trajectories were sorted based on whether they occurred before or after green (561 nm; PAPA) or violet (405 nm; DR) reactivation pulses, and state array analysis (*Heckert et al., 2022b*) was applied to each set of trajectories. For *Figure 3C*, reactivation by PAPA and DR for each cell were quantified by calculating the difference in total number of localizations before and after reactivation pulses.

## Statistical analysis

To derive a measure of error for SMT and SMT-PAPA, we performed bootstrapping analysis on all SMT datasets. For each dataset, a random sample of size n, where n is the total number of cells in the dataset, was drawn 100 times. The mean and standard deviation (stdev) from this analysis is reported.

Two-tailed p-values were calculated based on a normal distribution (Scipy function, scipy.stats.norm.sf) with mean equal to the difference between sample means and variance equal to the sum of the variances from bootstrap sampling (for SMT and SMT-PAPA) or biological replicates (for absolute abundance calculation of molecules from flow cytometry assay). The Šidák correction was applied to correct for multiple hypothesis testing within each experiment.

## Acknowledgements

We would like to thank all members of the Tjian-Darzacq lab for helpful discussions and suggestions over the years, in particular Claudio Cattaglio for advice on biochemistry experiments and Vinson Fan for suggestions on SMT data analysis. We are grateful to John J Ferrie for aiding with Rosetta modeling as well as both him and Jonathan P Karr for critical reading of the manuscript. We also thank Luke Lavis for providing fluorescent HaloTag ligands, CRL Flow Cytometry Facility for use of their instruments, the QB3 High Throughput Screening Facility for providing access to the Opera Phenix automated confocal microscope. This work was supported by the NIH grant U54-CA231641-01659 (to XD) and the Howard Hughes Medical Institute (to RT). TG was supported by a postdoctoral fellowship from the Jane Coffin Childs for Medical Research. AH was supported by the NIH Stem Cell Biological Engineering predoctoral fellowship T32 GM098218.

## Additional information

### Competing interests

Thomas GW Graham: is an inventor of pending patent application (PCT/US2021/062616) related to the use of PAPA as a molecular proximity sensor. Alec Heckert: is currently an employee of Eikon Therapeutics. Xavier Darzacq: is a co-founder of Eikon Therapeutics, Inc and an inventor on a pending patent application (PCT/US2021/062616) related to the use of PAPA as a molecular proximity sensor. The other authors declare that no competing interests exist.

### Funding

| Funder | Grant reference number | Author |
|---|---|---|
| National Institutes of Health | U54-CA231641-01659 | Xavier Darzacq |
| Howard Hughes Medical Institute | | Robert Tjian |

| Funder | Grant reference number | Author |
|---|---|---|
| Jane Coffin Childs Memorial Fund for Medical Research | | Thomas GW Graham |
| National Institutes of Health | T32 GM098218 | Alec Heckert |

The funders had no role in study design, data collection and interpretation, or the decision to submit the work for publication.

### Author contributions

Liza Dahal, Conceptualization, Data curation, Formal analysis, Validation, Investigation, Visualization, Methodology, Writing – original draft, Project administration, Writing – review and editing; Thomas GW Graham, Data curation, Formal analysis, Visualization, Methodology, Writing – review and editing, Performed and analyzed SMT-PAPA experiments; Gina M Dailey, Resources; Alec Heckert, Resources, Software; Robert Tjian, Xavier Darzacq, Supervision, Funding acquisition, Writing – review and editing

### Author ORCIDs

Liza Dahal ![ORCID] https://orcid.org/0000-0002-5600-1673
Thomas GW Graham ![ORCID] https://orcid.org/0000-0001-5189-4313
Gina M Dailey ![ORCID] https://orcid.org/0000-0002-8988-963X
Alec Heckert ![ORCID] https://orcid.org/0000-0001-8748-6645
Xavier Darzacq ![ORCID] https://orcid.org/0000-0003-2537-8395

Reviewer #1 (Public review): https://doi.org/10.7554/eLife.92979.3.sa1
Reviewer #2 (Public review): https://doi.org/10.7554/eLife.92979.3.sa2
Reviewer #3 (Public review): https://doi.org/10.7554/eLife.92979.3.sa3
Author response https://doi.org/10.7554/eLife.92979.3.sa4

## Additional files

### Supplementary files
MDAR checklist

### Data availability
Original western blots included in this study are available in accompanying source data zip files. SMT raw data for Figure 1-3 are accessible through https://doi.org/10.5281/zenodo.14009060.

The following dataset was generated:

| Author(s) | Year | Dataset title | Dataset URL | Database and Identifier |
|---|---|---|---|---|
| Dahal L | 2024 | Surprising Features of Nuclear Receptor Interaction Networks Revealed by Live Cell Single Molecule Imaging [Data set] | https://doi.org/10.5281/zenodo.14009060 | Zenodo, 10.5281/zenodo.14009060 |

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

## Appendix 1

### Brief overview of the steps involved in a PAPA-SMT experiment

A PAPA-SMT (*Graham et al., 2022*) experiment involves several steps (*Figure 3A*): (1) A SNAP-tagged target protein is labeled with a 'receiver' fluorophore such as JFX650, while a Halo-tagged partner protein is labeled with a 'sender' fluorophore such as JFX549. (2) The JFX650 receiver is placed in a dark state by illumination with intense red (639 nm) light. (3) A pulse of green (561 nm) light is used to excite the JFX549 sender, reactivating nearby dark-state JFX650 molecules. Because association of the two proteins brings the sender and receiver together, this step selectively reactivates complexes of the two proteins. (4) After switching off the green light, red light is used to image reactivated receiver molecules. (5) As an internal control, a violet (405 nm) light pulse is used to reactivate the receiver by 'DR', independent of proximity with the sender (*Heilemann et al., 2008*). (6) After switching off the violet light, reactivated molecules are again imaged with red light. Steps 3–6 are repeated for multiple rounds, and SMT trajectories occurring after green and violet pulses are analyzed separately using SASPT. Trajectories occurring after a green pulse (PAPA trajectories) are enriched for complexes between the SNAP- and Halo-tagged proteins, while trajectories occurring after a violet pulse (DR trajectories) approximate a random sample from the whole population.

## Appendix 2

**Appendix 2—table 1.** T2NRs expressed in U2OS cells according to the Expression Atlas (*Papatheodorou et al., 2020*; *Prakash et al., 2023*) and the Human Protein Atlas Databank (*Thul et al., 2017*; *Uhlén et al., 2015*).

| Name | TPM (Expression Atlas) | TPM (Human Protein Atlas) |
|---|---|---|
| RARα (NR1B2) | 43 | 47.1 |
| RARβ (NR1B2) | 0.6 | 0.1 |
| RARγ (NR1B3) | 39 | 16.5 |
| RXRα (NR2B1) | 28 | 23.4 |
| RXRβ (NR2B2)* | 56 | 5.3 |
| RXRγ (NR2B3) | n/a | 0 |
| THRA (NR1A1) | 41 | 18.2 |
| THRB (NR1A2) | 2 | 3 |
| LXRA (NR1H3) | 9 | 6.4 |
| LXRB (NR1H2) | 50 | 44.8 |
| COUP-TF1 (NR2F2) | 3 | 2.4 |
| COUP-TF2 (NR2F2) | 34 | 39.7 |
| VDR (NR1I1) | 9 | 7.3 |
| FXR (NR1H4) | n/a | 0.2 |
| PPARA | 14 | 7.7 |
| PPARD | 35 | 17.2 |
| PPARG | 7 | 11.4 |
| NURR (NRA42) | 2 | 0.9 |
| PXR (NR1I2) | 0.6 | 0 |
| CAR (NR1I3) | 26 | 0.3 |

TPM = transcripts per kilobase million, TPM values reported by the European Bioinformatics Institute Expression Atlas (ebi.ac.uk) and Human Protein Data Bank (proteinatlas.org) for Type II nuclear receptors (T2NRs) expression in U2OS cells.

*Note: We also observe expression of RXRβ in U2OS cells when probed with RXR antibody specific to the C-terminal epitope (Proteintech, #212181AP) in western blot (*Figure 1—figure supplement 2B*).

**Appendix 2—table 2.** Plasmid constructs used for co-immunoprecipitation and to make stable cell lines.

| Name | Promoter | Gene product | Short name in the paper | Appeared in |
|---|---|---|---|---|
| PB EF1α RARα-GDGAGLIN-Halo-3xFLAG IRES Puro | EF1α | RARα C-terminally fused with 3xFLAG-Halo tag through a short peptide linker sequence (GDGAGLIN) | RARα-Halo | *Figure 2A–C, Figure 2—figure supplement 1A–C, Figure 1—figure supplement 4B, Figure 2—figure supplement 3A* |
| PB EF1α V5-Halo-GDGAGLIN-RXRα IRES Puro | EF1α | RXRα N-terminally fused with V5-Halo tag through a short peptide linker sequence (GDGAGLIN) | Halo-RXRα | *Figure 2A–C, Figure 1—figure supplement 1B, Figure 2—figure supplement 1A–C, Figure 2—figure supplement 3A, Figure 1—figure supplement 4B* |
| PB EF1αRARα_M379R_T386R-GDGAGLIN-Halo-3xF IRES Puro | EF1α | RARRRα C-terminally fused with 3xFLAG-Halo tag through a short peptide linker sequence (GDGAGLIN) | RARαRR-Halo | *Figure 2—figure supplement 2A* |
| PB Puro EF1α-RARα-GDGAGLIN-3xFLAG | EF1a | RARα C-terminally fused with 3xFLAG tag through a short peptide linker sequence (GDGAGLIN) | RARα | *Figure 1—figure supplement 1B* |
| PB EF1a V5-RXRα IRES Puro | EF1α | RXRα N-terminally fused with V5 tag through a short peptide linker sequence (GDGAGLIN) | RXRα | *Figure 1—figure supplement 1B, Figure 2—figure supplement 2A* |

*Appendix 2—table 2 Continued on next page*

*Appendix 2—table 2 Continued*

| Name | Promoter | Gene product | Short name in the paper | Appeared in |
|---|---|---|---|---|
| PB EF1α ExMCS-GDGAGLIN-SNAPf-3xFLAG IRES Puro | EF1α | SNAPf-3xFLAG tag with a short peptide linker sequence (GDGAGLIN) | SNAP | *Figure 2D–F, Figure 2—figure supplement 2B–D, Figure 2—figure supplement 3B and C, Figure 3—figure supplement 1A and B* |
| PB EF1α RARα-GDGAGLIN-SNAPf-3xFLAG IRES Puro | EF1α | RARα C-terminally fused with 3xFLAG-SNAPf tag through a short peptide linker sequence (GDGAGLIN) | RARα-SNAP | *Figure 2D–F, Figure 3B and C, Figure 2—figure supplement 2B–D, Figure 2—figure supplement 3B and C, Figure 2—figure supplement 5A, Figure 3—figure supplement 1A and B* |
| PB EF1α RAR^RRα -GDGAGLIN-SNAPf-3xFLAG IRES Puro | EF1α | RAR^RRα C-terminally fused with 3xFLAG-SNAPf tag through a short peptide linker sequence (GDGAGLIN) | RARα^RR-SNAP | *Figure 2D–F, Figure 3B and C, Figure 2—figure supplement 2B–D, Figure 2—figure supplement 3B and C, Figure 2—figure supplement 5A, Figure 3—figure supplement 1A and B* |
| PB EF1α V5-SNAPf-GDGAGLIN-RXRα IRES Neo | EF1α | RXRα N-terminally fused with V5-SNAPf tag through a short peptide linker sequence (GDGAGLIN) | SNAP-RXRα | *Figure 2D–F, Figure 3B and C, Figure 2—figure supplement 2B–D, Figure 2—figure supplement 3B and C, Figure 3—figure supplement 1A and B* |
| PB PGK Puro EF1α SNAPf- 3xNLS | EF1α | SNAPf - 3xNLS linked through a short linker peptide sequence (GDGAGLIN) | SNAP-NLS | *Figure 3, Figure 3—figure supplement 1A and B* |

**Appendix 2—table 3.** Summary of single-molecule tracking (SMT) analysis for all conditions.

| Name | $f_{bound}$ | stdev | Num of cells | Trajs* | Appeared in |
|---|---|---|---|---|---|
| H2B-Halo | 75.5 | 0.7 | 22 | 108955 | *Figure 1C and D* |
| Halo-3xNLS | 11 | 0.9 | 22 | 67555 | *Figure 1C and D* |
| I RARα.c156 | 49 | 1 | 22 | 100148 | *Figure 1C and D, Figure 2C and F Figure 1—figure supplement 4A and B, Figure 2—figure supplement 4A and B* |
| I RARα.c239 | 47 | 1 | 22 | 42405 | *Figure 1C and D, Figure 2C* |
| I RARα.c258 | 53 | 1 | 22 | 74195 | *Figure 1C and D, Figure 2C* |
| I RXRα.c10 | 36 | 1 | 20 | 29690 | *Figure 1C and D, Figure 2C* |
| I RXRα.D6 | 36 | 1 | 20 | 24886 | *Figure 1C and D, Figure 2C and F, Figure 1—figure supplement 4A, Figure 2—figure supplement 4A and B, Figure 2—figure supplement 5A* |
| I RXRα.D9 | 42 | 1 | 20 | 30594 | *Figure 1C and D, Figure 2C* |
| I RARα.c156 w/o atRA | 53 | 1 | 22 | 14679 | *Figure 1—figure supplement 3A* |
| I RARα.c156 w/ 1 nM atRA | 54 | 1 | 22 | 15096 | *Figure 1—figure supplement 3A* |
| I RARα.c156 w/ 100 nM atRA | 53 | 1 | 22 | 13795 | *Figure 1—figure supplement 3A* |
| I RXRα.D6 w/o atRA | 34 | 1 | 20 | 17495 | *Figure 1—figure supplement 3A* |
| I RARα.D6 w/ 1 nM atRA | 31 | 1 | 20 | 17358 | *Figure 1—figure supplement 3A* |
| I RARα.D6 w/ 100 nM atRA | 31 | 1 | 20 | 18899 | *Figure 1—figure supplement 3A* |
| OE RARα-Halo | 27 | 0.72 | 22 | 21997 | *Figure 2C, Figure 1—figure supplement 4B, Figure 2—figure supplement 1B and C* |
| OE Halo-RXRα | 11 | 0.75 | 22 | 62767 | *Figure 2C, Figure 2—figure supplement 1B and C* |
| I RARα w/ SNAP | 47 | 1 | 22 | 35396 | *Figure 2F, Figure 2—figure supplement 4A and B* |
| I RARα w/ RARα^RR SNAP | 43 | 4 | 43 | 46435 | *Figure 2F, Figure 2—figure supplement 4A and B* |
| I RARα w/ SNAP-RXRα | 49 | 1 | 22 | 41411 | *Figure 2F, Figure 2—figure supplement 4A and B* |
| I RARα w/ RARα-SNAP | 29 | 5 | 46 | 34378 | *Figure 2F, Figure 2—figure supplement 4A and B* |
| I RXRα w/ SNAP | 38 | 1 | 22 | 26031 | *Figure 2F, Figure 2—figure supplement 4A and B* |
| I RXRα w/ RARα^RR-SNAP | 35 | 1 | 20 | 24137 | *Figure 2F, Figure 2—figure supplement 4A and B* |
| I RXRα w/ SNAP-RXRα | 16 | 2 | 22 | 24910 | *Figure 2F, Figure 2—figure supplement 4A and B* |
| I RXRα w/ RARα-SNAP | 56 | 1 | 22 | 25373 | *Figure 2F, Figure 2—figure supplement 4A nd B* |

*Trajs refers to total number of trajectories of the whole set.

**Appendix 2—table 4.** Number of molecules estimated according to the previously published protocol (*Cattoglio et al., 2019*).

| Name | Est. # molecules | Appeared in |
|---|---|---|
| Negative control | 0 (fixed) | *Figure 2—figure supplement 1A* <br> *Figure 2—figure supplement 2B* |
| CTCF | 109,800 (standard) | *Figure 2—figure supplement 1A* <br> *Figure 2—figure supplement 2B* |
| I RARα.c156 | 57,024±7048 | *Figure 2B* (replicate 1) and *Figure 2E* <br> *Figure 2—figure supplement 1A* |
| I RARα.c239 | 51,596±6144 | *Figure 2B* (replicate 2) |
| I RARα.c258 | 56,516±5260 | *Figure 2B* (replicate 3) |
| I RXRα.c10 | 53,548±4042 | *Figure 2B* (replicate 1) |
| I RXRα.D6 | 50,702±5616 | *Figure 2B* (replicate 2) and *Figure 2E* <br> *Figure 2—figure supplement 1A* |
| I RXRα.D9 | 50,946±5645 | *Figure 2B* (replicate 3) |
| I RARα (average) | 55,045±2997 | *Figure 2B* |
| I RXRα (average) | 51,732±1577 | *Figure 2B* |
| I RARα.c156 w/o atRA | 52,586±2230 | *Figure 1—figure supplement 3A* |
| I RARα.c156 w/ 1 nM atRA | 39,609±3180 | *Figure 1—figure supplement 3A* |
| I RARα.c156 w/ 100 nM atRA | 33,593±4673 | *Figure 1—figure supplement 3A* |
| I RXRα.D6 w/o atRA | 47,130±3291 | *Figure 1—figure supplement 3A* |
| I RXRα.D6 w/ 1 nM atRA | 45,849±660 | *Figure 1—figure supplement 3A* |
| I RXRα.D6 w/ 100 nM atRA | 43,021±1946 | *Figure 1—figure supplement 3A* |
| OE RARα-Halo | 233,732±3738 | *Figure 2B* <br> *Figure 2—figure supplement 1A* |
| OE Halo-RXRα | 905,005±1464 | *Figure 2B* <br> *Figure 2—figure supplement 1A* |
| I RARα w/ SNAP | 65,617±1293 | *Figure 2F* <br> *Figure 2—figure supplement 2B* |
| I RARα w/ RARα$^{RR}$-SNAP | 52,532±835 | *Figure 2F* <br> *Figure 2—figure supplement 2B* |
| I RARα w/ SNAP-RXRα | 62,392±1532 | *Figure 2F* <br> *Figure 2—figure supplement 2B* |
| I RARα w/ RARα-SNAP | 37,255±371 | *Figure 2F* <br> *Figure 2—figure supplement 2B* |
| I RXRα w/ SNAP | 50,898±1791 | *Figure 2F* <br> *Figure 2—figure supplement 2B* |
| I RXRα w/ RARα$^{RR}$-SNAP | 45,148±106 | *Figure 2F* <br> *Figure 2—figure supplement 2B* |
| I RXRα w/ SNAP-RXRα | 42,248±528 | *Figure 2F* <br> *Figure 2—figure supplement 2B* |
| I RXRα w/ RARα-SNAP | 58,880±1462 | *Figure 2F* <br> *Figure 2—figure supplement 2B* |

Errors were calculated from three biological replicates.

