## [Editor Report · eLife Assessment]

This **important** study provides data that challenges the standard model that binding of Type 2 Nuclear Receptors to chromatin is limited by the available pool of their common heterodimerization partner Retinoid X Receptor. The evidence supporting the conclusions is **compelling**, utilizing state-of-the-art single-molecule microscopy. This work will be of broad interest to cell biologists who wish to determine limiting factors in gene regulatory networks.

---

## [Referee Report · Reviewer #1 (Public review)]

This study provides compelling evidence that RAR, rather than its obligate dimerization partner RXR, is functionally limiting for chromatin binding. This manuscript provides a paradigm for how to dissect the complicated regulatory networks formed by dimerizing transcription factor families.

Dahal and colleagues use advanced SMT techniques to revisit the role of RXR in DNA-binding of the type-2 nuclear receptor (T2NR) RAR. The dominant consensus model for regulated DNA binding of T2NRs poses that they compete for a limited pool of RXR to form an obligate T2NR-RXR dimer. Using advanced SMT and proximity-assisted photoactivation technologies Dahal et al. now test the effect of manipulating the endogenous pool size of RAR and RXR on heterodimerization and DNA-binding in live U2OS cells. Surprisingly, it turns out that RAR, rather than RXR, is functionally limiting for heterodimerization and chromatin binding. By inference, the relative pool size of various T2NRs expressed in a given cell, rather than RXR, is likely determine chromatin binding and transcriptional output.

The conclusions of this study are well supported by the experimental results and provides unexpected novel insights in the functioning of the clinically important class of T2NR TFs. Moreover, the presented results show how the use of novel technologies can put long-standing theories on how transcription factors work upside down. This manuscript provides a paradigm for how to further dissect the complicated regulatory networks formed by T2NRs or other dimerizing TFs. I am convinced by the revised manuscript and have no additional concerns or comments.

---

## [Referee Report · Reviewer #2 (Public review)]

Summary:

In the manuscript "Surprising Features of Nuclear Receptor Interaction Networks Revealed by Live Cell Single Molecule Imaging", Dahal et al combine fast single molecule tracking (SMT) with proximity-assisted photoactivation (PAPA) to study the interaction between RARa and RXRa. The prevalent model in the nuclear receptor field suggests that type II nuclear receptors compete for a limiting pool of their partner RXRa. Contrary to this, the authors find that over-expression of RARa but not RXRa increases the fraction of RXRa molecules bound to chromatin, which leads them to conclude that the limiting factor is the abundance of RARa and not RXRa. The authors also perform experiments with a known RARa agonist, all trans retinoic acid (atRA) which has little effect on the bound fraction. Using PAPA, they show that chromatin binding increases upon dimerization of RARa and RXRa.

The authors have done well to address my comments and specify limitations where they could not.

---

## [Referee Report · Reviewer #3 (Public review)]

Summary:

This study aims to investigate the stoichiometric effect between core factors and partners forming the heterodimeric transcription factor network in living cells at endogenous expression levels. Using state-of-the-art single-molecule analysis techniques, the authors tracked individual RARα and RXRα molecules labeled by HALO-tag knock-in. They discovered an asymmetric response to the overexpression of counter-partners. Specifically, the fact that an increase in RARα did not lead to an increase in RXRα chromatin binding is incompatible with the previous competitive core model. Furthermore, by using a technique that visualizes only molecules proximal to partners, they directly linked transcription factor heterodimerization to chromatin binding.

Strengths:

The carefully designed experiments, from knock-in cell constructions to single-molecule imaging analysis, strengthen the evidence of the stoichiometric perturbation response of endogenous proteins. The novel finding that RXR, previously thought to be a target of competition among partners, is in excess provides new insight into key factors in dimerization network regulation. By combining the cutting-edge single-molecule imaging analysis with the technique for detecting interactions developed by the authors' group, they have directly illustrated the relationship between the physical interactions of dimeric transcription factors and chromatin binding. This has enabled interaction analysis in live cells that was challenging in single-molecule imaging, proving it is a powerful tool for studying endogenous proteins.

Weaknesses:

None noted.

---

## [Author Response]

The following is the authors’ response to the original reviews.

**Reviewer #1 (Public Review):**
This study provides compelling evidence that RAR, rather than its obligate dimerization partner RXR, is functionally limiting for chromatin binding. This manuscript provides a paradigm for how to dissect the complicated regulatory networks formed by dimerizing transcription factor families.Dahal and colleagues use advanced SMT techniques to revisit the role of RXR in DNA-binding of the type-2 nuclear receptor (T2NR) RAR. The dominant consensus model for regulated DNA binding of T2NRs posits that they compete for a limited pool of RXR to form an obligate T2NR-RXR dimer. Using advanced SMT and proximity-assisted photoactivation technologies, Dahal et al. now test the effect of manipulating the endogenous pool size of RAR and RXR on heterodimerization and DNA-binding in live U2OS cells. Surprisingly, it turns out that RAR, rather than RXR, is functionally limiting for heterodimerization and chromatin binding. By inference, the relative pool size of various T2NRs expressed in a given cell, rather than RXR, is likely to determine chromatin binding and transcriptional output.The conclusions of this study are well supported by the experimental results and provide unexpected novel insights into the functioning of the clinically important class of T2NR TFs. Moreover, the presented results show how the use of novel technologies can put long-standing theories on how transcription factors work upside down. This manuscript provides a paradigm for how to further dissect the complicated regulatory networks formed by T2NRs or other dimerizing TFs. I found this to be a complete story that does not require additional experimental work. However, I do have some suggestions for the authors to consider.
**Reviewer #1 (Recommendations For The Authors):**
(1) Does the increased chromatin binding measured when the RAR levels are increased reflect a higher occupancy of a similar set of loci, or are additional loci bound? The authors could discuss this issue in the context of the published literature. Obviously, this could be addressed experimentally by ChIP-seq or a similar analysis, but this would extend beyond the main topic of this manuscript.

We attempted to explore this experimentally using ChIP-seq with multiple RAR- and RXR-specific antibodies. Unfortunately, our results were inconclusive, as the antibody enrichment relative to the IgG control was insufficient for reliable interpretation. Specifically, our ChIP-seq enrichment levels were only around 1.5fold, while the accepted standard for meaningful ChIP enrichment is typically at least 2-fold. Due to these technical limitations, we decided to defer these experiments for now.

However, we agree with the reviewer that understanding whether the increased chromatin binding of RAR reflects higher occupancy at the same set of loci or binding to additional loci is a key question. In similar experiments involving the transcription factor TFEB (Esbin et al., 2024, *Genes Dev*, doi: 10.1101/gad.351633.124) where an increase in the SMT bound fraction occurred, both scenarios—higher occupancy at known loci and binding to additional loci in ChIP-seq was observed. So, addressing this intriguing possibility in future studies focused on RAR and RXR would be interesting.

(2) The results presented suggest convincingly that endogenous RXR is normally in excess to its binding partners (in U2OS cells). This point could be strengthened further by reducing RXR levels, e.g., by knocking out 1 allele or the use of shRNAs (although the latter method might be too hard to control). Overexpression of another T2NR might also help determine the buffer capacity of RXR.

We appreciate the reviewers’ acknowledgment that our results convincingly demonstrate that endogenous RXR is typically in excess relative to its binding partners in U2OS cells. We agree that this conclusion could be further reinforced by experiments such as overexpression of another T2NR to test RXR's buffering capacity. We are actively pursuing follow-up experiments involving overexpression of additional T2NRs to address this question in more detail. These studies are ongoing, and we plan to explore the buffer capacity of RXR more extensively in a future manuscript.

(3) The ~10% difference in fbound of RAR and RXR (in Figs 1 and 2), while they should be 1:1 dimers, is explained by invoking the expression of RXR isoforms. Can the authors be more specific concerning the nature of these isoforms?

We have provided detailed information about different T2NRs expressed in U2OS cells according to the Expression Atlas and the Human Protein Atlas Database in Supplementary Table S1. Table S1 specifically shows that both isoforms of RXRα and RXRβ are expressed in U2OS cells. Additionally, the caption of Table S1 explicitly notes the presence of isoform RXRβ in U2OS cells. In the main text, we reference Table S1 when discussing the 10% difference in fbound between RARα and RXRα, and we have now suggested that the expression of RXRβ likely accounts for the observed discrepancy.

**Reviewer #2 (Public Review):**
Summary:In the manuscript "Surprising Features of Nuclear Receptor Interaction Networks Revealed by Live Cell Single Molecule Imaging", Dahal et al combine fast single molecule tracking (SMT) with proximity-assisted photoactivation (PAPA) to study the interaction between RARa and RXRa. The prevalent model in the nuclear receptor field suggests that type II nuclear receptors compete for a limiting pool of their partner RXRa. Contrary to this, the authors find that over-expression of RARa but not RXRa increases the fraction of RXRa molecules bound to chromatin, which leads them to conclude that the limiting factor is the abundance of RARa and not RXRa. The authors also perform experiments with a known RARa agonist, all trans retinoic acid (atRA) which has little effect on the bound fraction. Using PAPA, they show that chromatin binding increases upon dimerization of RARa and RXRa.Strengths:In my view, the biggest strength of this study is the use of endogenously tagged RARa and RXRa cell lines. As the authors point out, most previous studies used either in vitro assays or over-expression. I commend the authors on the generation of single-cell clones of knock-in RARa-Halo and Halo-RXRa. The authors then carefully measure the abundance of each protein using FACS, which is very helpful when comparing across conditions. The manuscript is generally well written and figures are easy to follow. The consistent color-scheme used throughout the manuscript is very helpful.Weaknesses:(1) Agonist treatment:The authors test the effect of all trans retinoic acid (atRA) on the bound fraction of RARa and RXRa and find that "These results are consistent with the classic model in which dimerization and chromatin binding of T2NRs are ligand independent." However, all the agonist treatments are done in media containing FBS. FBS is not chemically defined and has been found to have between 10 and 50 nM atRA (see references in PMID 32359651 for example). The addition of 1 nM or 100 nM atRA is unlikely to result in a strong effect since the medium already contains comparable or higher levels of agonist. To test their hypothesis of ligand-independent dimerization, the authors should deplete the media of atRA by growing the cells in a medium containing charcoal-stripped FBS for at least 24 hours before adding agonist.

We acknowledge the reviewer's concern regarding the presence of atRA in FBS and agree that it may introduce baseline levels of agonist. However, in our experiments, both the 1 nM and 100 nM atRA treatments resulted in observable changes in RAR expression levels (Figure S3C). Additionally, the luciferase assays demonstrated that 100 nM atRA significantly increased retinoic acid-responsive promoter activity (Figure S1C). Given these clear responses to atRA, we believe the observed lack of effect on the chromatin-bound fraction cannot be attributed to the presence of comparable or higher levels of atRA in the FBS, as the reviewer suggests. Moreover, since our results align with the established literature and do not impact the core findings of our study, we decided not to pursue the suggested experiments with charcoal-stripped FBS in this manuscript.

(2) Photobleaching and its effect on bound fraction measurements:The authors discard the first 500 to 1000 frames due to the high localization density in the initial frames. This will preferentially discard bound molecules that will bleach in the initial frames of the movie and lead to an over-estimation of the unbound fraction.For experiments with over-expression of RAR-Halo and Halo-RXR, the authors state that the cells were pre-bleached and that these frames were used to calculate the mean intensity of the nuclei. When pre-bleaching, bound molecules will preferentially bleach before the diffusing population. This will again lead to an over-representation of the unbound fraction since this is the population that will remain relatively unaffected by the pre-bleaching. Indeed, the bound fraction for over-expressed RARa and RXRa is significantly lower than that for the corresponding knock in lines. To confirm whether this is a biological result, I suggest that the authors either reduce the amount of dye they use so that this pre-bleaching is not necessary or use the direct reactivation strategy they use for their PAPA experiments to eliminate the pre-bleaching step.As for the measurement of the nuclear intensity, since the authors have access to multiple HaloTag dyes, they can saturate the HaloTagged proteins with a high concentration of JF646 or JFX650 to measure the mean intensity of the protein while still using the PA-JFX549 for SMT. Together, these will eliminate the need to prebleach or discard any frames.

The Janelia Fluor dyes used in our experiments are known for their high photostability (Grimm et al., 2021, *JACS Au*, doi: 10.1021/jacsau.1c00006). During the initial 80 ms imaging to calculate the mean nuclear intensity, the laser power was kept at very low intensity (~3%) for a brief duration (~10 seconds), in contrast to the high-intensity (~100%) used during the tracking experiments, which span around 3 minutes. This low-power illumination does not induce significant photobleaching but merely puts the dyes in a temporary dark state. Therefore, this pre-bleaching step closely resembles the direct reactivation strategy employed in our PAPA experiments.

To further address the reviewer's concern, we performed a frame cut-off analysis for our SMT movies of endogenous RARα-Halo and over-expressed RARα-Halo (Figure S9B). The analysis shows no significant change in the bound fraction of either endogenous or over-expressed RARα-Halo when discarding the initial 1000 frames. Based on these results, we conclude that the pre-bleaching does not lead to an overestimation of the unbound fraction, and that our experimental approach is robust.

(3) Heterogeneous expression of the SNAP fusion proteins:The cell lines expressing SNAP tagged transgenes shown in Fig S6 have very heterogeneous expression of the SNAP proteins. While the bulk measurements done by Western blotting are useful, while doing single-cell experiments (especially with small numbers - ~20 - of cells), it is important to control for expression levels. Since these transgenic stable lines were not FACS sorted, it would be helpful for the reader to know the spread in the distribution of mean intensities of the SNAP proteins for the cells that the SMT data are presented for. This step is crucial while claiming the absence of an effect upon over-expression and can easily be done with a SNAPTag ligand such as SF650 using the procedure outlined for the over-expressed HaloTag proteins.

We agree with the reviewer that there is heterogeneity in SNAP protein expression across the transgenic lines. In response to the reviewer’s suggestion, we performed the proposed experiment to assess the distribution of mean intensities for two key experimental conditions: Halo-RXRα with overexpressed RARα-SNAP and HaloRXRα with overexpressed RARαRR-SNAP. These results again confirm that the increase in chromatin-bound fraction of Halo-RXRα is observed only in the presence of RARα capable of heterodimerizing with RXRα, supporting our main conclusion (Figure S9).

For these experiments, we followed the same labelling procedure described in the methods section for tracking endogenous Halo-tagged proteins alongside transgenic SNAP proteins. As shown in Figure S9, for ~ 70 cell nuclei, the distribution of mean intensities is similar for both conditions, with the bound fraction of Halo-RXRα significantly increasing in the presence of RARα-SNAP compared to RARαRR-SNAP. This analysis underscores that the observed effects are indeed due to the functional differences between the two RARα variants rather than variability in expression levels.

(4) Definition of bound molecules:The authors state that molecules with a diffusion coefficient less than 0.15 um2/s are considered bound and those between 1-15 um2/s are considered unbound. Clarification is needed on how this threshold was determined. In previous publications using saSPT, the authors have used a cutoff of 0.1 um2/s (for example, PMID 36066004, 36322456). Do the results rely on a specific cutoff? A diffusion coefficient by itself is only a useful measure of normal diffusion. Bound molecules are unlikely to be undergoing Brownian motion, but the state array method implemented here does not seem to account for non-normal diffusive modes. How valid is this assumption here?

We acknowledge the inconsistency in the diffusion coefficient thresholds for defining the chromatin-bound fraction used across our group’s publications. The choice of threshold or cutoff (0.1 µm²/s vs 0.15 µm²/s) is largely arbitrary and does not significantly impact the results. To validate this, we tested the effect of different cutoffs on fbound (%) for endogenously expressed Halo-tagged RARα and RXRα (Figure S10). As shown in Figure S10, there was no substantial difference in fbound (%) calculated using a 0.1 µm²/s versus 0.15 µm²/s cutoff (e.g., RARα clone c156: 47±1% vs 49±1%; RXRα clone D6: 34±1% vs 35±1%).

Since we have consistently applied the 0.15 µm²/s cutoff throughout this manuscript across all experimental conditions, the comparative analysis of fbound (%) remains valid. While we agree that a Brownian diffusion model may not fully capture the motion of bound molecules, our state array model accounts for localization error, which likely incorporates some of the chromatin motion features. Moreover, the distinction between bound (<0.15 µm²/s) and unbound (1-15 µm²/s) populations is sufficiently large that using a normal diffusion model is reasonable for our analysis.

(5) Movies:Since this is an imaging manuscript, I request the authors to provide representative movies for all the presented conditions. This is an essential component for a reader to evaluate the data and for them to benchmark their own images if they are to try to reproduce these findings.

We have now included representative movies for all the SMT experimental conditions presented in the manuscript. Please see data availability section of the manuscript.

(6) Definition of an ROI:The authors state that "ROI of random size but with maximum possible area was selected to fit into the interior of the nuclei" while imaging. However, the readout speed of the Andor iXon Ultra 897 depends on the size of the defined ROI. If the ROI was variable for every movie, how do the authors ensure the same sampling rate?

We used the frame transfer mode on the Andor iXon Ultra 897 camera for our acquisitions, which allows for fast frame rate measurements without altering the exposure time between frames. Additionally, we verified the metadata of all our movies to ensure a consistent frame interval of 7.4 ms across all conditions. This confirms that the sampling rate was maintained uniformly, despite the variability in ROI size.

**Reviewer #2 (Recommendations For The Authors):**
(1) 'Hoechst' is mis-spelled.

We have now corrected this typo in the manuscript.

(2) Cos7 appears in several places throughout the text. I assume this is a typo. If so, please correct it. If not, please explain if some experiments were done in Cos7 cells and kindly provide a justification for that.

The use of Cos7 cells is intentional and not a typo. Cos7 cells have been previously utilized in studies investigating the interaction between T2NRs (Kliewer et al., 1992, *Nature*, doi: 10.1038/355446a0). In our study, due to technical issues with antibodies for coIP in U2OS cells, we initially used Cos7 cells for control experiments to verify that Halo-tagging of RARα and RXRα did not disrupt their interaction, by transiently expressing the constructs in Cos7 cells. Following these control experiments, we confirmed the direct interaction of endogenously expressed RAR and RXR in U2OS cells with their respective binding partners using the SMT-PAPA assay. Since these results confirmed that Halo-tagging did not interfere with RAR-RXR interactions, we chose not to repeat the coIP experiments in U2OS cells.

**Reviewer #3 (Public Review):**
Summary:This study aims to investigate the stoichiometric effect between core factors and partners forming the heterodimeric transcription factor network in living cells at endogenous expression levels. Using state-of-the-art single-molecule analysis techniques, the authors tracked individual RARα and RXRα molecules labeled by HALO-tag knock-in. They discovered an asymmetric response to the overexpression of counter-partners. Specifically, the fact that an increase in RARα did not lead to an increase in RXRα chromatin binding is incompatible with the previous competitive core model. Furthermore, by using a technique that visualizes only molecules proximal to partners, they directly linked transcription factor heterodimerization to chromatin binding.Strengths:The carefully designed experiments, from knock-in cell constructions to singlemolecule imaging analysis, strengthen the evidence of the stoichiometric perturbation response of endogenous proteins. The novel finding that RXR, previously thought to be a target of competition among partners, is in excess provides new insight into key factors in dimerization network regulation. By combining the cutting-edge single-molecule imaging analysis with the technique for detecting interactions developed by the authors' group, they have directly illustrated the relationship between the physical interactions of dimeric transcription factors and chromatin binding. This has enabled interaction analysis in live cells that was challenging in single-molecule imaging, proving it is a powerful tool for studying endogenous proteins.Weaknesses:As the authors have mentioned, they have not investigated the effects of other T2NRs or RXR isoforms. These invisible factors leave room for interpretation regarding the origin of chromatin binding of endogenous proteins (Recommendations 4). In the PAPA experiments, overexpressed factors are visualized, but changes in chromatin binding of endogenous proteins due to interactions with the overexpressed proteins have not been investigated. This might be tested by reversing the fluorescent ligands for the Sender and Receiver. Additionally, the PAPA experiments are likely to be strengthened by control experiments (Recommendations 5).

We agree that this would be an interesting experiment. However, there are three technical challenges that complicate its implementation: First, as demonstrated in our original PAPA paper, dark state formation is less efficient when dyes are conjugated to Halo compared to SNAPf, making the reverse configuration less optimal. Second, SNAPf-tagged proteins have slower labeling kinetics than Halotagged proteins, often resulting in under-labeling of SNAPf. Third, our SNAPf transgenes were integrated polyclonally. Since background PAPA scales with the concentration of the sender-labeled protein, variable concentrations of the senderlabeled SNAPf proteins would introduce significant variability, complicating the interpretation of the background PAPA signal. Due to these concerns, we believe that performing reciprocal measurements with reversed fluorescent ligands may not yield reliable results.

**Reviewer #3 (Recommendations For The Authors):**
(1) The term "Surprising features" in the title is ambiguous and may force readers to search for what it specifically refers to. Including a word that evokes specific features might be helpful.

Our findings contradict previous work, which suggested that chromatin binding of T2NRs is regulated by competition for a limited pool of RXR. In contrast, we found that RAR expression can limit RXR chromatin binding, but not the other way around, which challenges the existing model. This unexpected result is what we refer to as a "surprising feature" in our title, and we believe it accurately reflects the novel insights our study provides. We also think that this is clearly conveyed in our manuscript abstract, supporting the use of "Surprising features" in the title.

(2) p.3, line 11 - The threshold of 0.15 μm2s-1 seems to be a crucial value directly linked to the value of fbound. What is the rationale for choosing this specific value? If consistent conclusions can be obtained using threshold values that are similar but different, it would strengthen the robustness of the results.

Please refer to our response to Reviewer #2’s Public Review point 4. The threshold choice is arbitrary and doesn’t affect the overall conclusions. To test this, we compared fbound (%) values calculated using both 0.1 μm²s-1 and 0.15 μm²s-1 cutoffs. For example, with endogenously expressed Halo-tagged RARα (clone c156), we observed fbound values of 47±1% vs 49±1%, and for RXRα (clone D6), 34±1% vs 35±1%, respectively (Figure S10). Since we have consistently applied the 0.15 μm²s-1 cutoff across all experimental conditions in this manuscript, the comparisons of fbound (%) between different conditions are robust and valid.

(3) p.4, line 13 - "the fbound of endogenous RARα-Halo (47{plus minus}1%) was largely unchanged upon expression of SNAP (47{plus minus}1%)" part of the sentence is not surprising. It would make more sense if it were expressed as "the fbound of endogenous RARα-Halo (47{plus minus}1%) was largely unchanged upon expression of RXRα-SNAP (49{plus minus}1%), consistent with the control SNAP (47{plus minus}1%).".

We understand how the original phrasing may be confusing to the readers and have restructured the sentence as suggested by the reviewer for clarity.

(4) p.6, line 26 - The discussion that "most chromatin binding of endogenous RXRα in U2OS cells depends on heterodimerization partners other than RARα" seems to contradict the top right figure in Figure 4. If that's the case, the binding partner for the bound red molecule might be yellow rather than blue. Given a decrease in the number of RARα molecules with an unchanged binding ratio, the total number of binding molecules has decreased. Could it be interpreted that the potential reduction in RXRα chromatin binding, accompanying the decrease in binding RARα, is compensated for by other partners?

We agree with the reviewer that both the yellow and blue molecules in Figure 4 represent T2NRs that can heterodimerize with RXR. For simplicity, we chose to omit the depiction of RXR dimerization with other T2NRs (represented in yellow) in Figure 4. We have now included a note in the figure caption to clarify this. We plan to follow up on the buffer capacity of RXR with other T2NRs in a separate manuscript and will discuss this aspect in more detail once we have data from those experiments.

(5) Fig. 3 - I expected that DR localizations always appear more frequently than PAPA localizations by the difference in the number of distal molecules. Why does the linear line for SNAP-RXRα in Fig. 3 B have a slope exceeding 1? Also, although the sublinearity is attributed to binding saturation, is there any possibility that this sublinearity originates from the PAPA system like the saturation of PAPA reactivation? Control samples like Halo-SNAPf-3xNLS might address these concerns.

The number of DR and PAPA localizations depends on the arbitrarily chosen intensity and duration of green and violet light pulses. For any given protein pair, different experimental settings can result in PAPA localizations being greater than, less than, or equal to the number of DR localizations. Therefore, the informative metric is not the absolute number of DR and PAPA localizations, but rather how the ratio of PAPA to DR localizations changes between different conditions—such as between interacting pairs and non-interacting controls.

Regarding the sublinearity, we agree that it is essential to consider whether the observed sublinearity might stem from saturation of the PAPA signal. We know of two ways in which this could occur:

First, PAPA can be saturated as the duration of the green light pulse increases and dark-state complexes are depleted. However, this cannot explain the nonlinearity that we observe, because the duration of the green light pulse is constant, and thus the probability that a given complex is reactivated by PAPA is also constant. Likewise, holding the violet pulse duration constant yields a constant probability that a given molecule is reactivated by DR. PAPA localizations are expected to scale linearly with the number of complexes, while DR localizations are expected to scale linearly with the total number of molecules. Sublinear scaling of PAPA localizations with DR localizations thus implies that the number of complexes scales sublinearly with the total concentration of the protein.

Second, saturation could occur if PAPA localizations are undercounted compared to DR localizations. While this is a valid concern, we consider it unlikely in this case because (1) our localization density is below the level at which our tracking algorithm typically undercounts localizations, and (2) we observe sublinearity for RXR → RAR PAPA even though the number of PAPA localizations is lower than the DR localizations; undercounting due to excessive localization density would be expected to introduce the opposite bias in this case.

(6) Fig. 4 - The differences between A, B, and C on the right side of the model are subtle, making it difficult to discern where to see. Emphasizing the difference in molecule numbers or grouping free molecules at the top might help clarify these distinctions.

We appreciate the reviewer’s feedback. In response, we have revised Figure 4 by grouping the free molecules on the top right side for panels A, B and C, as suggested.

(7) While the main results are obtained through single-molecule imaging, no singlemolecule fluorescence images or trajectory plots are provided. Even just for representative conditions, these could serve as a guide for readers trying to reproduce the experiments with different custom-build microscope setups. Also, considering data availability, depositing the source data might be necessary, at least for the diffusion spectra.

We have now included representative movies for all the presented SMT conditions as source data. Please see data availability section of the manuscript.

(8) Tick lines are not visible on many of the graph axes.

We have revised the figures to ensure that the tick lines are now clearly visible on all graph axes.

(9) Inconsistencies in the formatting are present in the methods, such as "hrs" vs. "hours", spacing between numbers and units, and "MgCl2". "u" should be "μ" and "x" should be "×".

We have corrected the formatting errors.

(10) Table S4, rows 16 and 17 - Are "RAR"s typos for "RXR"s?

We have corrected this in the manuscript.

(11) p.10~12 - Are three "Hoestch"s typos for "Hoechst"s?

This is now corrected in the manuscript.

(12) p.11, line 17 - According to the referenced paper, the abbreviation should be "HILO" in all capital letters, not "HiLO".

This is now corrected in the manuscript.

(13) "%" on p.3, line 18, and "." on p.6, line 27 are missing.

This missing “%” and “.” are now added.